# The M-current works in tandem with the persistent sodium current to set the speed of locomotion

**Jérémy Verneuil[1], Cécile Brocard[1], Virginie Trouplin[1], Laurent Villard[2], Julie Peyronnet-Roux[1‡], Frédéric Brocard[1‡]***

**1** Institut de Neurosciences de la Timone, Aix-Marseille Université and Centre National de la Recherche Scientifique, Marseille, France, **2** Aix-Marseille Université, Inserm, MMG, Marseille, France

‡ These authors are joint senior authors on and contributed equally to this work.
* frederic.brocard@univ-amu.fr

**Data Availability Statement:** All relevant data are within the paper and its Supporting information files.

## Abstract

The central pattern generator (CPG) for locomotion is a set of pacemaker neurons endowed with inherent bursting driven by the persistent sodium current ($I_{NaP}$). How they proceed to regulate the locomotor rhythm remained unknown. Here, in neonatal rodents, we identified a persistent potassium current critical in regulating pacemakers and locomotion speed. This current recapitulates features of the M-current ($I_M$): a subthreshold noninactivating outward current blocked by 10,10-bis(4-pyridinylmethyl)-9(10H)-anthracenone dihydrochloride (XE991) and enhanced by N-(2-chloro-5-pyrimidinyl)-3,4-difluorobenzamide (ICA73). Immunostaining and mutant mice highlight an important role of Kv7.2-containing channels in mediating $I_M$. Pharmacological modulation of $I_M$ regulates the emergence and the frequency regime of both pacemaker and CPG activities and controls the speed of locomotion. Computational models captured these results and showed how an interplay between $I_M$ and $I_{NaP}$ endows the locomotor CPG with rhythmogenic properties. Overall, this study provides fundamental insights into how $I_M$ and $I_{NaP}$ work in tandem to set the speed of locomotion.

## Introduction

Locomotion requires a recurrent activation of muscles with variable rhythm to adapt speed of movements as circumstances demand. In mammals, rhythmicity appears to be ensured by a network mainly localized in the ventromedial gray matter of upper lumbar segments [1, 2]. The rhythm-generating network is a set of pacemaker cells endowed with intrinsic bursting activity in a frequency range similar to stepping rhythms [3, 4]. In exploring the ionic basis for rhythmogenesis, we identified the persistent sodium current ($I_{NaP}$) as a critical current in burst-generating mechanism [5–8]. The immediate assumption was that the locomotor rhythm may emerge from neurons incorporating $I_{NaP}$ as a "pacemaker" current. In line with this concept, inhibition of $I_{NaP}$ abolishes locomotor-like activity in rodents [3, 9–11] and salamanders [12] and disrupts locomotion in zebrafish [8, 13] and *Xenopus laevis* tadpoles [14]. Altogether, a picture emerges that the locomotor rhythm arises from a dynamic interplay

**Funding:** This research was financed by grants from French National Research Agency (CalpaSCI, ANR-16-CE16-0004; IMprove, ANR-19-CE17-0018) and the French Institut pour la Recherche sur la Moelle épinière et l'Encéphale (IRME). The funders had no role in study design, data collection and analysis, decision to publish, or preparation of the manuscript.

**Competing interests:** The authors have declared that no competing interests exist.

**Abbreviations:** aCSF, artificial cerebrospinal fluid; AIS, axonal initial segment; CHO, Chinese hamster ovary; CNS, central nervous system; CPG, central pattern generator; ES, embryonic stem; FBS, Fetal Bovine Serum; f–I, frequency–intensity relationship; GFP, green fluorescent protein; ICA73, *N*-(2-chloro-5-pyrimidinyl)-3,4-difluorobenzamide; IR-DIC, infrared differential interference contrast; I–V, current–voltage; mEPSC, miniature excitatory postsynaptic current; $Na_v$, voltage-gated sodium channel; NGS, Natural Goat Serum; NMA, N-methyl-DL aspartate; TEA, tetraethylammonium; TTX, tetrodotoxin; XE991, 10,10-bis(4-pyridinylmethyl)-9(10H)-anthracenone dihydrochloride; 5-HT, 5-hydroxytryptamine.

between circuit-based activity and pacemaker burst-generating mechanisms with a critical role of $I_{NaP}$ for the initiation of bursts [4, 10].

Among general principles of rhythmogenesis, outward conductances are required to repolarize bursts [15]. In vertebrates, the $Ca^{2+}$-activated $K^+$ current ($I_{KCa}$) appears important in repolarizing bursts to regulate the locomotor rhythm [16–19], among other hyperpolarizing currents mediated by Kv1.2 [20] and A-type $K^+$ channels [21] or $Na^+/K^+$ pumps [22–24]. Considering $I_{NaP}$ as a subthreshold persistent conductance critically engaged in the burst initiation, it might be efficiently counteracted by a subthreshold persistent potassium current to end bursts of pacemakers and thereby to regulate the locomotor cycle. The Kv7 (KCNQ) channels mediate a unique subthreshold persistent outward current named $I_M$ [25]. The Kv7 channels are widely expressed in the central nervous system (CNS), and several pieces of evidence reveal the existence of $I_M$ in spinal motoneurons, presumably mediated by heteromeric Kv7.2/Kv7.3 channels [26–30]. However, their expression at the level of the locomotor central pattern generator (CPG) and their functional role in locomotion remain unexplored.

The present study characterizes the functional expression of $I_M$ in locomotor-related interneurons and identifies Kv7.2-containing channels as its molecular constituent. We show that the modulation of $I_M$ adjusts $I_{NaP}$ amplitude and regulates the emergence of pacemaker properties, their burst repolarization, the frequency regime of the locomotor CPG, and the speed of locomotion. In sum, we provide the first description of $I_M$ in a vertebrate motor CPG, to our knowledge, and describe its dynamic interplay with $I_{NaP}$ as a fundamental mechanism in shaping bursting activity of neurons and networks that control rhythmic locomotor output.

## Results

### M-current regulates the speed of locomotion

To study the role of $I_M$ in the locomotor behavior of juvenile rats (15–21 days old), potent and selective Kv7 channel modulators were injected intraperitoneally at a concentration commonly used for a systemic modulation of $I_M$ [31–33]. The regular pattern of walking was captured with the CatWalk system before and after 30 min of drug administration (Fig 1). Control experiments with DMSO discarded potential effects of the vehicle ($P > 0.05$, in gray, Fig 1A–1C). Rats walked faster when the broad-spectrum activation of Kv7.2–Kv7.5 channels occurred by injecting retigabine (5 mg/kg; see [34]), which was related to a shorter stance phase ($P < 0.05$, in green, Fig 1A–1C). The broad-spectrum inhibition of Kv7 channels by linopirdine (3 mg/kg; [35]) had opposite effects, i.e., rats walked slower, attributable to a longer stance phase ($P < 0.05$, in yellow, Fig 1A–1C). A second set of experiments focused on the role of Kv7.2/3 subunits by means of more selective drugs. The potent Kv7.2/3 channel opener *N*-(2-chloro-5-pyrimidinyl)-3,4-difluorobenzamide (ICA73) (5 mg/kg; [36]) ($P < 0.05$, in blue, Fig 1A–1C) and the Kv7.2/3 channel blocker 10,10-bis(4-pyridinylmethyl)-9(10H)-anthracenone dihydrochloride (XE991) (5 mg/kg; [35]) ($P < 0.05$, in red, Fig 1A–1C) reproduced retigabine and linopirdine effects, respectively. Noteworthy, none of the Kv7 modulators affected the interlimb coordination or gait ($P > 0.05$; S1A–S1D Fig).

Modulation of the step cycle through systemic administration of drugs may involve changes in the excitability of midbrain circuits setting the speed of locomotion [37, 38]. To investigate a putative modulation of $I_M$ at the level of the spinal locomotor CPG, the opener ICA73 (0.05 mg/kg) and the blocker XE991 (0.025 mg/kg) were injected intrathecally at the L1–L2 level in adult rats. The intrathecal delivery of drugs replicated the effects observed with an intraperitoneal injection. Specifically, rats walked faster with ICA73 ($P < 0.05$, in blue, Fig 1D–1F) and slower with XE991 ($P < 0.05$, in red, Fig 1D–1F) without affecting interlimb coordination or

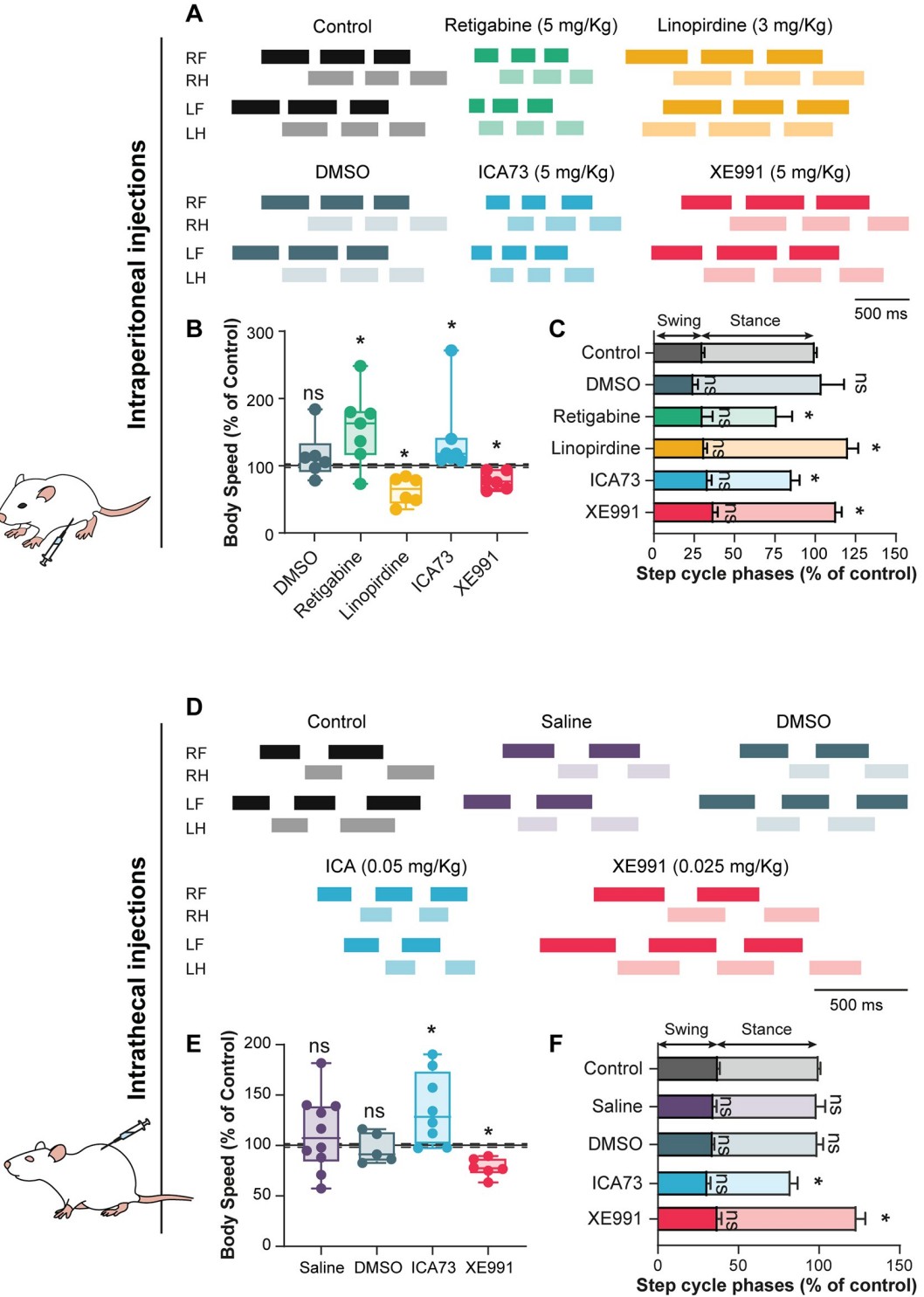

**Fig 1. $I_M$ sets the speed of locomotion.** (A,D) Representative footfall diagrams during CatWalk locomotion of juvenile rats (A, 15–21 days old) before (black) and 30 min after acute i.p. administration of DMSO (gray, $n = 6$ rats), retigabine (5 mg/kg, green, $n = 7$ rats), linopirdine (3 mg/kg, yellow, $n = 6$ rats), ICA73 (5 mg/kg, blue, $n = 7$ rats), or XE991 (5 mg/kg, red, $n = 6$ rats) or in adult rats (D) before (black) and 5–10 min after acute i.t. administration at the L1–L2 level of saline (purple, $n = 10$ rats), DMSO (gray, $n = 5$ rats), ICA73 (0.05 mg/kg, blue, $n = 8$ rats), or XE991 (0.025 mg/kg, red, $n = 6$ rats). The stance phase is indicated by horizontal bars and the swing phase by open spaces. (B,E) Normalized changes of the body speed induced in juvenile (B) and adult (E) rats by the abovementioned drugs. Dotted lines indicate the 95% confidence intervals of control

values. $^*P < 0.05$, comparing animals before and after drug administration; Wilcoxon paired test. (C,F) Normalized changes of swing and stance phases induced in juvenile (C) and adult (F) rats by the abovementioned drugs and expressed as a percentage of the total step cycle. $^*P < 0.05$, comparing data collected before and after drug administration; Wilcoxon paired test. Data in C and F are mean ± SEM. Underlying numerical values can be found in the S1 Data. ICA73, *N*-(2-chloro-5-pyrimidinyl)-3,4-difluorobenzamide; i.p., intraperitoneal; i.t., intrathecal; LF, left forelimb; LH, left hindlimb; ns, not significant; RF, right forelimb; RH, right hindlimb; XE991, 10,10-bis(4-pyridinylmethyl)-9(10H)-anthracenone dihydrochloride.

gait ($P > 0.05$; S1E–S1H Fig). Note that control experiments with DMSO or saline discarded potential effects of vehicles ($P > 0.05$, in gray and in purple, Fig 1D–1F).

Altogether, these results support the concept that $I_M$, presumably mediated by Kv7.2 and/or Kv7.3 channels, is an important and physiologically relevant regulator of the spinal locomotor network without interfering with the dynamic postural control.

## Ubiquitous expression of Kv7.2-containing channels in interneurons from the locomotor CPG region

We then studied the expression of Kv7.2/3 channels in ventromedial interneurons from upper lumbar segments (L1–L2), where components of the locomotor rhythm-generating network are located [1, 2]. Immunostaining substantiates the presence of Kv7.2 channels in the soma of all ventromedial interneurons and in the distal part of their axonal initial segments (AISs), identified by their expression of voltage-gated sodium (Na$_v$) channels (Fig 2A–2C, 2G, and 2H). A similar Kv7.2-immunostaining profile was observed in lumbar motoneurons (S2A–S2C, S2G, and S2H Fig). Conversely to Kv7.2, the Kv7.3 subunit was weakly expressed in the soma and virtually undetectable in AISs of most ventromedial interneurons (Fig 2D–2G). On the other hand, Kv7.3 immunostaining distinctively labeled the distal part of AISs in almost half of lumbar motoneurons (S2D–S2H Fig). In sum, L1–L2 ventromedial interneurons ubiquitously express Kv7.2-containing channels, suggestive of the existence of $I_M$ in the locomotor CPG. Note that we validated the specificity the antibodies against Kv7.2 and Kv7.3 on Chinese hamster ovary (CHO) cells stably expressing Kv7.2 or Kv7.3 channels (S3 Fig).

## Characterization of $I_M$ in the rhythm-generating kernel for locomotion

To assess whether interneurons from the CPG region exhibit a functional $I_M$, we isolated $I_M$ by whole-cell patch-clamp recordings under tetrodotoxin (TTX) (1 μM). By using the standard relaxation protocol, we measured $I_M$ as the amplitude of the tail inward current evoked by stepping down voltage from −10 mV (Fig 3A). All L1–L2 ventromedial neurons displayed an electrophysiological signature of $I_M$ [39]. From the current–voltage (*I*–*V*) relationship fitted with a standard Boltzmann function (in black, Fig 3B), the threshold for activation ($V_T$) was positive to −67.3 ± 1.9 mV, and its amplitude increased steeply (slope factor k: 4.4 ± 0.6) for larger voltage steps with a midpoint of activation ($V_{1/2}$) at −43.6 ± 1.5 mV and then plateaued above −30 mV. The peak amplitude of $I_M$ was in the mean of 79.2 ± 7.4 pA. We further characterized $I_M$ pharmacologically (Fig 3C). The $I_M$ enhancer ICA73 (10 μM) increased the holding current and the magnitude of $I_M$ and hyperpolarized its $V_T$ and $V_{1/2}$ activation ($P < 0.05$, in blue, Fig 3B–3D). The $I_M$ enhancer retigabine (100 nM) replicated the effects of ICA73 on $I_M$ (S4A–S4F Fig). The $I_M$ blocker XE991 (10 μM) markedly reduced the holding current and virtually abolished $I_M$ ($P < 0.05$, in red, Fig 3B–3D). Note that similar biophysical properties of $I_M$, sensitive to XE991, were recorded from transgenic mice Hb9:green fluorescent protein (GFP)-positive ventromedial interneurons that are considered as components of the locomotor rhythm-generating circuit (S5 Fig) [40, 41]. Given the strong expression of Kv7.2 in interneurons, we evaluated the contribution of Kv7.2 to $I_M$ with knock-in mice bearing the T274M

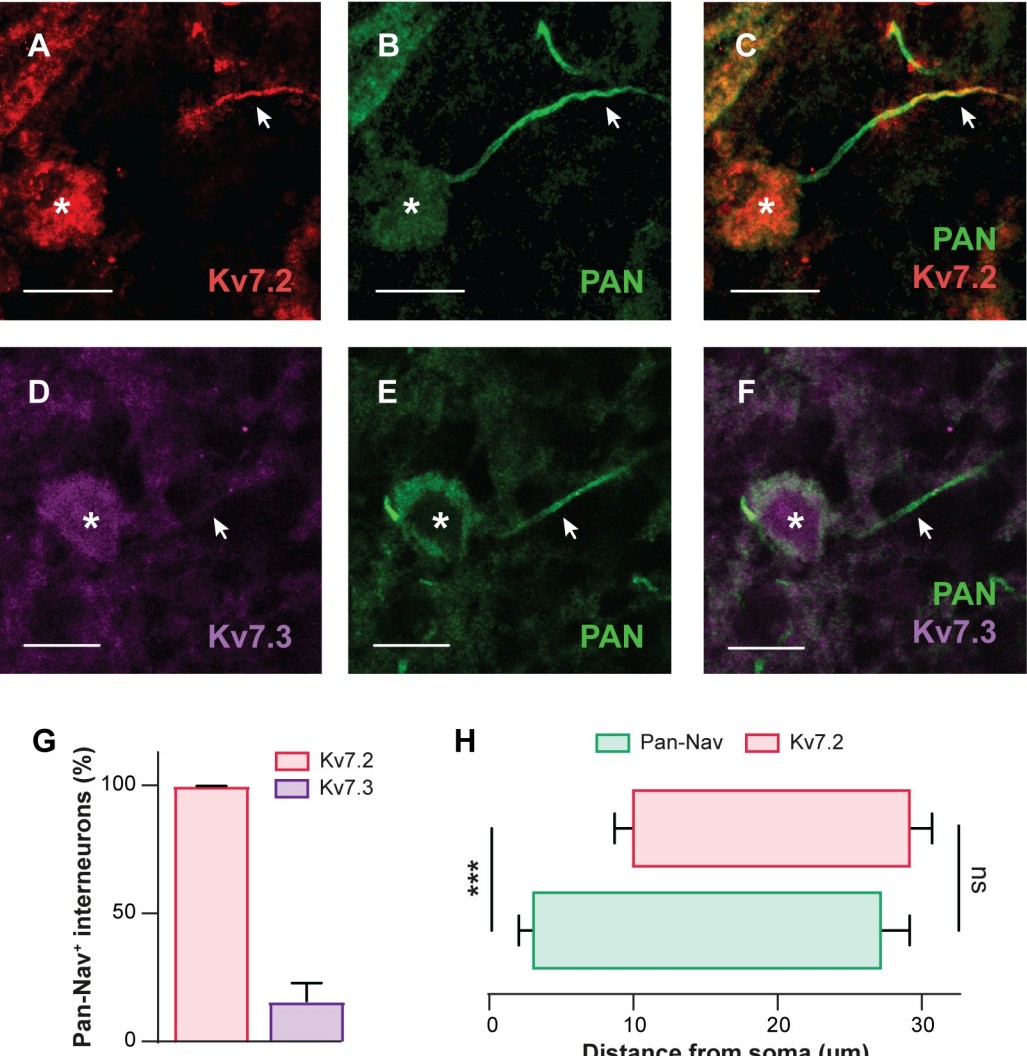

**Fig 2. Lumbar interneurons within the locomotor CPG region ubiquitously express Kv7.2-containing channels.** (A–F) Immunostaining of lumbar (L1–L2) ventromedial interneurons from juvenile rats ($n$ = 3 rats) against Kv7.2 (A, $n$ = 216 cells) or Kv7.3 (D, $n$ = 170 cells) along the AIS labeled by the pan-Na$_v$ antibody (B,E). Kv7.2 and pan-Na$_v$ are merged in (C), and Kv7.3 and pan-Na$_v$ are merged in (F). Asterisks indicate the nucleus position and arrowheads the AIS. Scale bars = 10 μm. (G) Group means quantification of the proportion of pan-Na$_v$ positive interneurons expressing Kv7.2 or Kv7.3 channels. (H) Group means quantification of the start and end positions of pan-Na$_v$ and Kv7.2 immunolabeling along the axonal process from the soma ($n$ = 10 cells). $^{***}P < 0.001$, comparing start or end positions between groups; Mann–Whitney test. Data are mean ± SEM. Underlying numerical values can be found in the S1 Data. AIS, axonal initial segment; CPG, central pattern generator; Na$_v$, voltage-gated sodium channel.

Kv7.2 mutation associated with Ohtahara syndrome [42]. Because this mutation is homozygous lethal, we studied $I_M$ on Kv7.2$^{Thr274Met/+}$ animals. We found in heterozygous Kv7.2$^{Thr274Met/+}$ mice that $I_M$ was about halved in amplitude relative to Kv7.2$^{+/+}$ littermates (Fig 3E). These results indicated that $I_M$ was at least mediated by K$_v$7.2-containing channels. To further investigate which Kv7 subunits are involved, we exploited the differential sensitivity of Kv7 channels to tetraethylammonium (TEA) [43–46]. Thus, Kv7.2 homomers are almost fully blocked at 1 mM, whereas K$_v$7.3, K$_v$7.4, and K$_v$7.5 homomers are blocked with IC$_{50}$ values of >200, approximately 3.0, and approximately 70 mM, respectively. K$_v$7.2/7.3 and K$_v$7.2/7.5

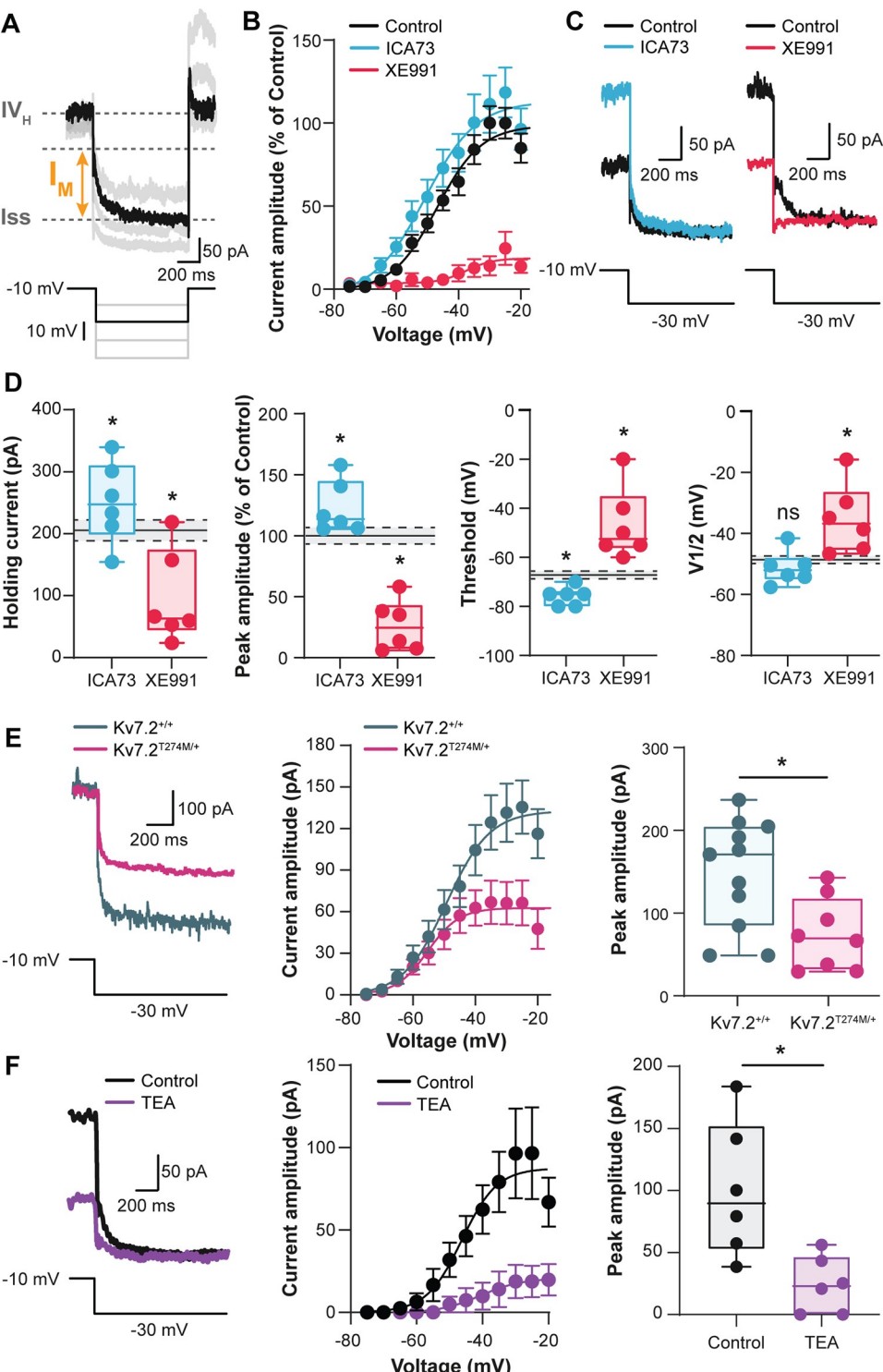

**Fig 3. Characterization of $I_M$ in interneurons of the locomotor CPG region.** (A) Representative inward current relaxation recorded in L1–L2 ventromedial interneurons from neonatal rats and induced by the standard $I_M$ deactivation voltage protocol. The current relaxation ($I_M$) was measured as the difference between the fast transient ($I_{VH}$) and the steady-state current (*Iss*). (B) Boltzmann-fitted *I–V* relationships of $I_M$ normalized to control (black) values and established before and after bath-applying XE991 (10 µM, red) or ICA73 (10 µM, blue). Error bars represent ± SEM. (C) Representative deactivation of $I_M$ before and after adding the abovementioned drugs. (D) Boxplots quantification of biophysical properties of $I_M$. (*n* = 6 cells). Dashed lines with gray shading indicate the 95%

confidence intervals of control values. $^*P < 0.05$, $^{***}P < 0.001$, comparing data collected before and after adding the abovementioned drugs; Wilcoxon paired test. (E) Representative deactivation (left), Boltzmann-fitted $I–V$ relationships (middle), and amplitude (right) of $I_M$ recorded in interneurons from wild-type ($n = 11$ cells, gray) and Kv7.2$^{Thr274Met/+}$ mutant mice ($n = 8$ cells, purple). $^*P < 0.05$; Mann–Whitney test. (F) Representative deactivation (left), Boltzmann-fitted $I–V$ relationships (middle), and amplitude (right) of $I_M$ recorded in interneurons from neonatal rat before and after adding TEA (1 mM, $n = 6$ cells). $^*P < 0.05$; Mann–Whitney test. Underlying numerical values can be found in the S1 Data. CPG, central pattern generator; ICA73, $N$-(2-chloro-5-pyrimidinyl)-3,4-difluorobenzamide; $I–V$, current–voltage; TEA, tetraethylammonium; XE991, 10,10-bis(4-pyridinylmethyl)-9(10H)-anthracenone dihydrochloride.

heteromeric channels show intermediate sensitivities to TEA block with IC$_{50}$ values near approximately 3 mM and approximately 200 mM. We found that bath application of 1 mM TEA blocked approximately 80% of $I_M$ in L1–L2 ventromedial interneurons from neonatal rats ($P < 0.05$, in purple, Fig 3F).

Altogether, these results support the expression of a noninactivating K$^+$ current corresponding to $I_M$ in the locomotor CPG, presumably carried by Kv7.2-containing channels mostly in a homomeric form.

## M-current sets the neuronal excitability and gates pacemaker bursting mode

We characterized the role of $I_m$ in membrane properties of L1–L2 ventromedial interneurons recorded from neonatal rats. As a first observation, the $I_m$ enhancer ICA73 hyperpolarized cells (in blue, S6A Fig; routinely compensated to −60 mV) along with a drop of the input resistance ($P < 0.05$ Table 1). Therefore, interneurons became less excitable (higher rheobase; $P < 0.05$, Table 1) and produced fewer spikes ($P < 0.05$, in blue, Fig 4A) without any changes in parameters of the action potential ($P > 0.05$, Table 1). The frequency–intensity (f–I) curve was thus shifted to the right (Fig 4B). The reversibility of these effects when the $I_m$ blocker XE991 was applied emphasized the dependence of ICA73 on Kv7 channels (S6A–S6D Fig). Consistent with this, the Kv7 opener retigabine reproduced effects of ICA73 on electroresponsive properties (S4G and S4H Fig and Table 1). We further tested the functional implication of $I_m$ by using the $I_m$ blocker XE991 alone. The $I_m$ blocker per se neither depolarized the resting membrane potential nor increased the neuronal excitability or the firing rate even 30 min after the bath application of the drug ($P > 0.05$, in red, Fig 4A and 4B and Table 1). Thus, the basal excitability was not affected by XE991. However, $I_m$ blockers such as XE991 have the peculiarity to be voltage-dependent blockers, with higher affinity at positive potentials, and therefore

**Table 1. Effects of drugs targeting Kv7 channels on passive and active membrane properties of interneurons.**

| | V$_{rest}$ (mV) | Input Resistance (MΩ) | Rheobase (pA) | Spike Threshold (mV) | Spike Amplitude (mV) |
|---|---|---|---|---|---|
| Control | −57.8 ± 1.2 | 1,147.3 ± 121.3 | 11.3 ± 1.4 | −45.0 ± 1.1 | 55.5 ± 2.2 |
| ICA73 ($n = 6$) | −65.4 ± 2.8* | 694.3 ± 222.4* | 28.3 ± 10.7* | −49.1 ± 1.4 (ns) | 48.1 ± 2.6 (ns) |
| Retigabine ($n = 9$) | −70.7 ± 1.7** | 869.4 ± 103.2* | 17.2 ± 3.8* | −47.1 ± 1.9 (ns) | 48.4 ± 1.8 (ns) |
| XE991 ($n = 7$) | −57.1 ± 1.8 (ns) | 979.1 ± 150.8 (ns) | 17.1 ± 2.6 (ns) | −43.7 ± 1.7 (ns) | 57.9 ± 4.0 (ns) |

Values are means ± SEM; $n$, number of cells. Mean firing frequency was measured at 2-fold the rheobase.

$^*P < 0.05$,

$^{***}P < 0.001$, comparing data collected before and after bath-applying drugs mentioned above; Wilcoxon paired test. Underlying numerical values can be found in the S1 Data.

**Abbreviations:** ICA73, $N$-(2-chloro-5-pyrimidinyl)-3,4-difluorobenzamide; ns, not significant; XE991, 10,10-bis(4-pyridinylmethyl)-9(10H)-anthracenone dihydrochloride.

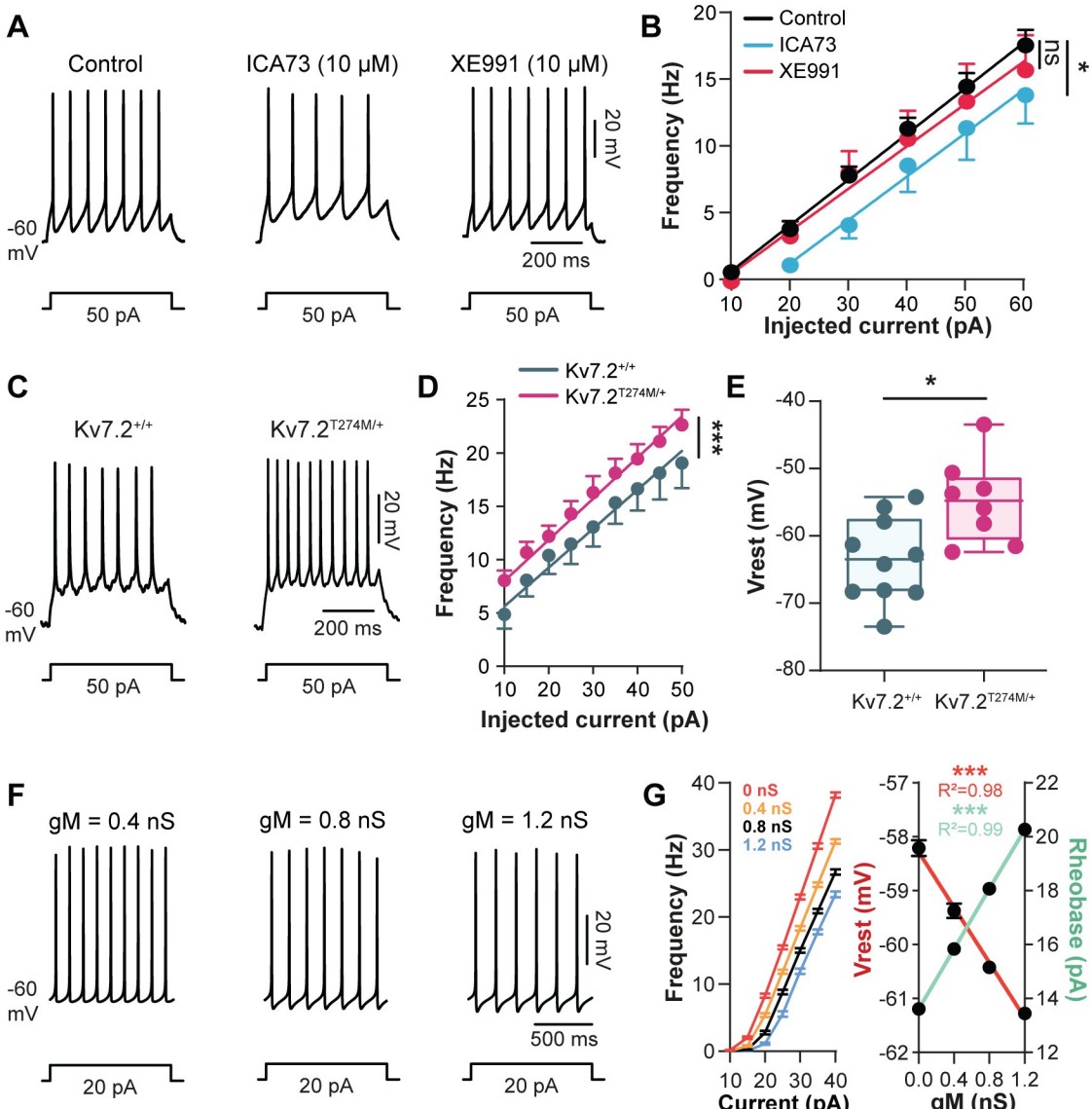

**Fig 4. $I_M$ mediated by Kv7.2-containing channels regulates both the excitability and the spiking activity of L1–L2 ventromedial interneurons.** (A,B) Spiking activity of ventromedial interneurons (L1–L2) to a near-threshold depolarizing pulse (A) with its respective frequency–current relationships (B) before (black) and after bath-applying ICA73 (10 μM, $n = 7$ cells, blue) or XE991 (10 μM, $n = 7$ cells, red). Continuous lines represent the best-fitting linear regression. $^{***}P < 0.001$ comparison of the fits before and after adding the abovementioned drugs. (C–E) Representative spiking activity (C), frequency–current relationships (D), and resting membrane potential (E) of ventromedial interneurons (L1–L2) recorded from wild-type ($n = 10$ cells) and Kv7.2$^{Thr274Met/+}$ mutant mice ($n = 8$ cells). Continuous lines represent the best-fitting linear regression. $^{***}P < 0.001$ comparison of the fits. $^{*}P < 0.05$, comparing resting membrane potentials; Mann–Whitney test. (F) Spiking activity from a single-neuron model at 3 different values of $g_M$. The injected current pulses in A, C, and F are indicated below voltage traces. (G) Mean plots of the firing rate, rheobase, and resting membrane potential as function of $g_M$ (in the heterogeneous population model of 50 neurons). The continuous line is the best-fitting linear regression. $^{***}P < 0.001$, Spearman correlation test. $R^2$ indicates the correlation index. Data are mean ± SEM. Underlying numerical values can be found in the S1 Data. ICA73, $N$-(2-chloro-5-pyrimidinyl)-3,4-difluorobenzamide; XE991, 10,10-bis(4-pyridinylmethyl)-9(10H)-anthracenone dihydrochloride.

are very poor inhibitors at perithreshold potentials [47]. Furthermore, the inhibition develops slowly upon depolarization within a timescale of minutes [48]. To overcome this experimental limitation, we used the Kv7.2$^{Thr274Met/+}$ mutant mice for which $I_m$ was halved (see above). We showed that L1–L2 ventromedial interneurons from mutant mice displayed a higher spiking

frequency ($P < 0.001$, Fig 4C and 4D) and a more depolarized resting membrane potential ($P < 0.05$, Fig 4E).

We also studied the theoretical effect of $I_m$ by modeling $I_m$ in an Hb9 cell model that we previously used [10]. The model was supplemented by $I_m$ derived from our voltage-clamp recordings. A population of 50 uncoupled interneurons was simulated with a randomized normal distribution of neuronal parameters (see Materials and methods). Simulated neurons reproduced key features of the biological responses to stepwise depolarizing currents with firing rate in the range of our experimental data (Fig 4F and 4G). The increase of the M-conductance ($g_M$) in the model qualitatively captured the modulation of neuronal excitability observed experimentally with $I_m$ enhancers ICA73 or retigabine; firing rate decreased, the resting membrane potential hyperpolarized, and the rheobase increased ($P < 0.001$, Fig 4F and 4G). However, in contrast to our electrophysiological recordings with the $I_m$ blocker XE991, decreasing $g_M$ in the model predicted a depolarization of $V_{rest}$, whereas the firing rate and the rheobase would go up and down, respectively ($P < 0.001$, Fig 4F and 4G). On the other hand, these computational data were similar to those recorded in Kv7.2$^{Thr274Met/+}$ mutant mice, making them suitable to use as a tool to explore how $I_m$ might shape the firing pattern of interneurons.

A remarkable effect of modeling a decrease of $g_M$ was the gradual transfer to a bursting mode in a small proportion of neurons to reach 17% of bursters when $g_M$ was switched off (Fig 5A and 5B). Bursts disappeared when $I_{NaP}$ was zeroed (Fig 5A). To evaluate this computational prediction, we tested XE991 on cells intracellularly recorded and constantly depolarized with a suprathreshold current (Fig 5C). In this condition, XE991 caused a transition from tonic spiking to bursting in approximately 18% of the interneurons recorded (Fig 5C and 5D). Bursts were abolished by the $I_{NaP}$ blocker riluzole (Fig 5C). The insensitivity of bursts to kynurenic acid (1.5 mM; blocker of the fast-glutamatergic transmission) and the lack of rhythmic currents in voltage-clamp recordings precluded a role of network inputs in the emergence of bursts (Fig 5C).

Overall, these data indicate an important role of $I_M$ in setting the excitability and firing properties of interneurons within the locomotor CPG region, notably by impeding the initiation of bursts mediated by $I_{NaP}$.

## Interneurons balance $I_M$ and $I_{NaP}$ to trigger pacemaker bursting mode

The emergence of $I_{NaP}$-dependent bursting cells when Kv7 channels are blocked suggests that $I_M$ might counteract $I_{NaP}$ to regulate pacemaker properties. To test this possibility, voltage-clamp recordings were performed to examine the degree of interaction between the 2 currents. In response to very slow voltage ramps, ventromedial interneurons from neonatal rats displayed a large inward current attributable to $I_{NaP}$ (Fig 6A; see [5]). In the presence of $I_M$ blocker XE991, $I_{NaP}$ was higher in amplitude, whereas $V_T$ and $V_{1/2}$ did not change ($P < 0.05$, Fig 6A and 6B). Similar results were obtained from Kv7.2$^{Thr274Met/+}$ mutant mice ($P < 0.05$, Fig 6C and 6D). These data show that biophysical properties of $I_M$ are well-suited to counteract the depolarizing drive furnished by $I_{NaP}$. Here, we speculate that the combination of $I_M$ and $I_{NaP}$ currents is a possible mechanism in controlling bursting dynamics. We evaluated this assumption by varying the maximal conductance levels of $g_M$ and $g_{NaP}$ in the heterogeneous population model composed of 50 uncoupled Hb9-type interneurons (Fig 6E). None of the neurons exhibited bursting at base levels of $g_M$ (0.8 nS) and $g_{NaP}$ (0.4 nS). We found that a population bursting activity could be triggered by either a reduction of $g_M$ or an increase of $g_{NaP}$ in turn (Fig 6E). However, the striking observation was the synergistic effect of reducing $g_M$ and increasing $g_{NaP}$ conductance on the generation of bursts ($P < 0.001$, Fig 6F). Compared with

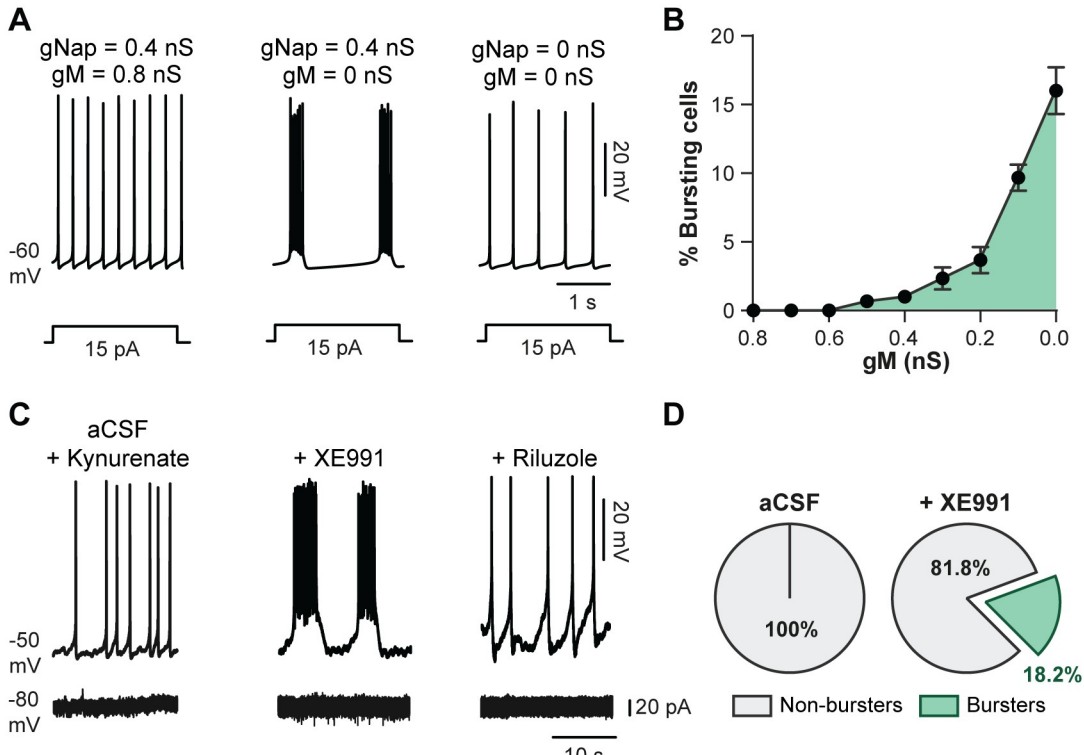

**Fig 5. $I_M$ regulates the firing pattern of L1–L2 ventromedial interneurons.** (A) Typical switch of the firing pattern from spiking to bursting in a single-neuron model when $g_M$ was switched "off"; simulated blockade of $I_{NaP}$ ($g_{NaP} = 0$) abolished bursting activity. (B) Dependence of the percentage of bursting cells on $g_M$ in the heterogeneous population model of 50 neurons. (C) On top, representative voltage traces of a ventromedial interneuron (L1–L2) intracellularly recorded in the presence of kynurenate (1.5 mM) before (left) and after (middle) bath-applying XE991 (10 μM). A subsequent application of riluzole (5 μM) abolished bursts (right). At bottom, current traces of the cell illustrated and voltage clamped at −80 mV. (D) Proportion of burster and nonburster interneurons before and after bath-applying XE991 (*n* = 22 cells). Data are mean ± SEM. Underlying numerical values can be found in the S1 Data. aCSF, artificial cerebrospinal fluid; XE991, 10,10-bis (4-pyridinylmethyl)-9(10H)-anthracenone dihydrochloride.

nonbursters, bursters were distinguished by a more negative $V_{1/2}$ of $I_{NaP}$ but displayed a similar $V_{1/2}$ of $I_M$ (*P* < 0.001, Fig 6G).

Taken together, these results suggest that most CPG interneurons are endowed with the intrinsic ability to switch from spiking to bursting behavior through a sliding balance between $I_{NaP}$ and $I_M$.

## M-current controls bursting dynamics of pacemaker cells

Our modeling study, combined with electrophysiological data, supports fine modulation of $g_M$ as a key mechanism for the emergence of pacemaker cells. Here, we used the model as a tool to delineate the role of $I_m$ in bursting dynamics. To study the dependence of bursting characteristics on $I_M$, we simulated a negative-voltage shift of $I_{NaP}$ activation ($V_{1/2} = −54$ mV) to convert a tonic cell into burster as a result of reducing $[Ca^{2+}]_o$ [10]. As previously described, the burst period and the burst duration decreased as the neuron was depolarized (Fig 7A). When $g_M$ was omitted, the burst duration as well as the interburst interval increased (*P* < 0.001, Fig 7B and 7C). Opposite effects on burst timing were observed when $g_M$ was increased (*P* < 0.001, Fig 7C). Notably, at high levels of $g_m$, a subset of bursting cells in the heterogeneous population model switched to tonic spiking (S7A and S7B Fig). To determine the dynamic contributions

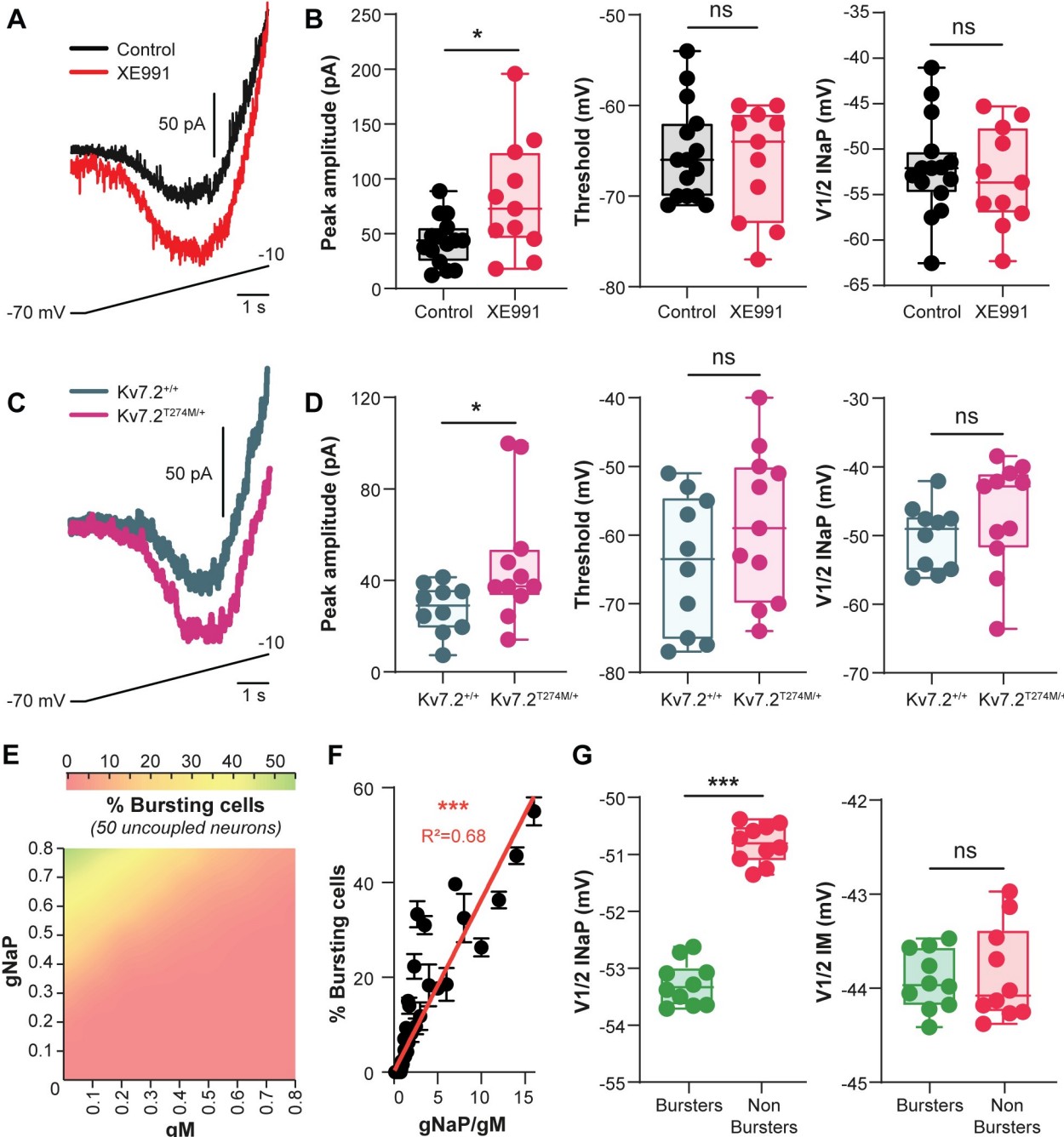

**Fig 6. A balance of $I_M$ and $I_{NaP}$ gates bursting mode of CPG interneurons.** (A,C) Raw traces of leak-subtracted persistent sodium current ($I_{NaP}$) evoked in L1–L2 ventromedial interneuron by a slow voltage ramp and recorded from neonatal rats (A) under control condition ($n$ = 15 cells, black) or in the presence of XE991 (10 μM, $n$ = 11 cells, red) or from neonatal mice (C) wild-type ($n$ = 10 cells, gray) and Kv7.2^Thr274Met/+ mutant mice ($n$ = 11 cells, pink). (B,D) Boxplot quantification of the biophysical properties of $I_{NaP}$. *$P$ < 0.05; Mann–Whitney test. (E) Dependence of the percentage of bursting pacemaker cells on $g_M$ and $g_{NaP}$ in the heterogeneous population model of 50 neurons. Results are pooled from 10 simulations. (F) Percentage of bursting cells as function of the $g_{NaP}/g_M$ ratio. Each dot represents average values of 10 simulations for a fixed $g_{NaP}/g_M$ ratio. Continuous line is the best-fitting linear regression. ***$P$ < 0.001, Spearman correlation test. $R^2$ indicates the correlation index. (G) Boxplot quantification of $V_{1/2}$ of $I_{NaP}$ and $I_M$ for burster and nonburster groups. Each dot represents 1 simulation in the heterogeneous population model of 50 neurons. Parameters values for $g_{NaP}$ and $g_M$ were set at 0.8 nS and 0.1 nS, respectively. ***$P$ < 0.001; Mann–Whitney test. Underlying numerical values can be found in the S1 Data. CPG, central pattern generator; XE991, 10,10-bis(4-pyridinylmethyl)-9(10H)-anthracenone dihydrochloride.

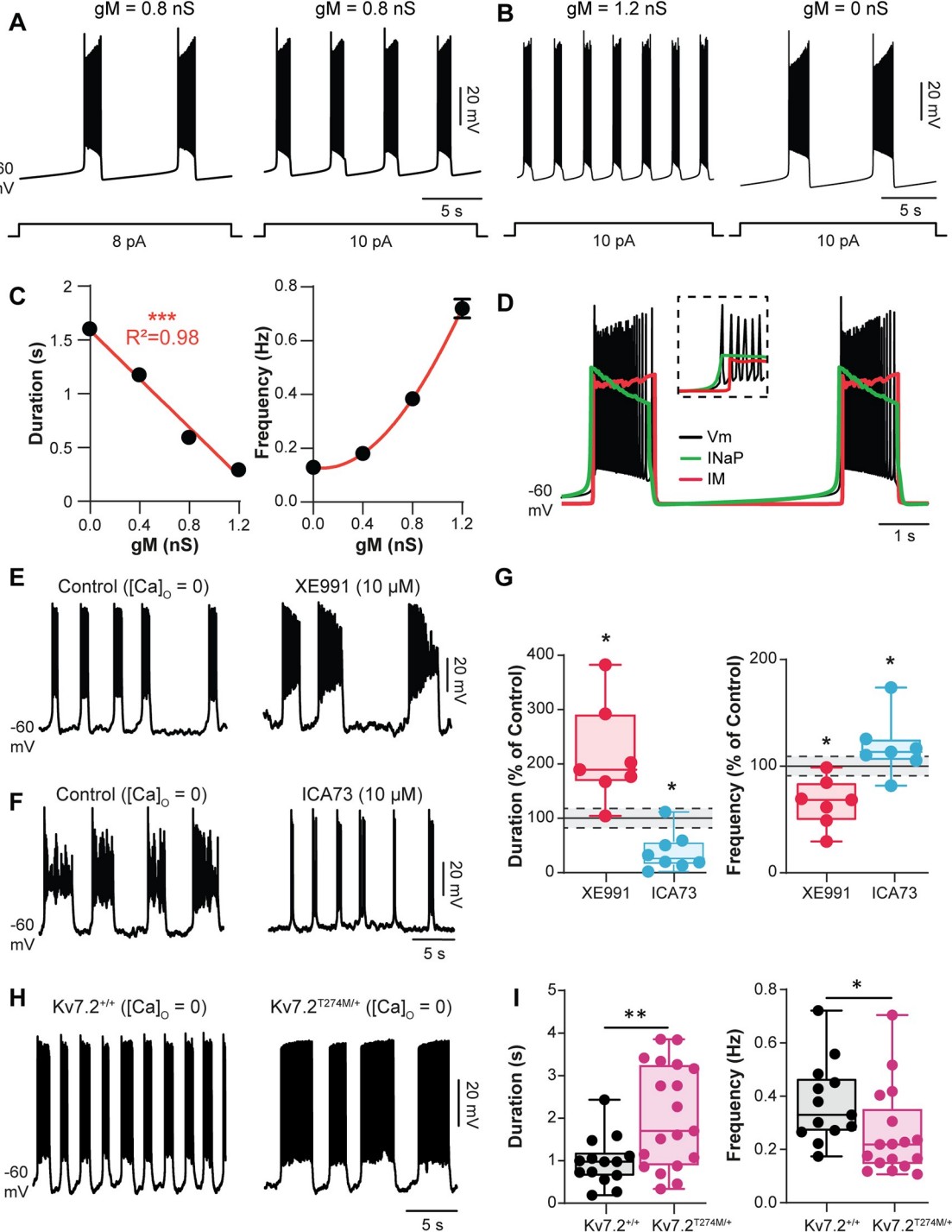

**Fig 7. $I_M$ controls burst dynamics.** (A,B) Firing behavior from a bursting pacemaker neuron model in response to incrementing depolarizing current injections (A) or for 2 different values of $g_M$ (B). The injected current pulses are indicated below voltage traces $V_{1/2}$ $I_{NaP}$ = −54 mV. (C) Duration and frequency of bursts as function of $g_M$ in the heterogeneous population model of 50 neurons. Results are pooled from 10 simulations. $R^2$ indicates the correlation index. (D) Dynamics of the neuronal membrane potential (black), the $I_{NaP}$ (green), and the $I_M$ (red) during bursting activity of a single pacemaker neuron model. The broken-line box highlights dynamics of $I_{NaP}$ and $I_M$ at the onset of bursts. (E,F) $[Ca^{2+}]_o$-free-saline–induced bursting activity recorded intracellularly from L1–L2 ventromedial interneurons before and after XE991 (10 μM, $n$ = 7 cells) (E) or ICA73 (10 μM, $n$ = 7 cells) (F). (G) Normalized changes of burst parameters. Dashed lines with gray shading indicate the 95% confidence intervals of control values. $^*P < 0.05$, comparing data before and after the abovementioned drugs; Wilcoxon paired test. (H) $[Ca^{2+}]_o$-free-saline–

induced bursting activity recorded intracellularly in L1–L2 ventromedial interneurons from wild-type ($n$ = 13 cells) and Kv7.2$^{Thr274Met/+}$ mutant mice ($n$ = 18 cells). (I) Boxplot quantification of burst parameters. $^*P < 0.05$, comparing wild-type versus mutant; Mann–Whitney test. Underlying numerical values can be found in the S1 Data. ICA73, $N$-(2-chloro-5-pyrimidinyl)-3,4-difluorobenzamide; XE991, 10,10-bis(4-pyridinylmethyl)-9(10H)-anthracenone dihydrochloride.

of $I_M$ and $I_{NaP}$ to the bursting activity, we examined changes of $g_M$ and $g_{NaP}$ at specific time points during the burst itself and during interburst intervals (Fig 7D). During the beginning of each burst, $g_{NaP}$ preceded the activation of $g_m$. Over the course of the burst, a slow decrease of $g_{NaP}$ was observed, whereas $g_m$ slightly increased. At the end of the burst, $g_m$ and $g_{NaP}$ relaxed to a baseline level over the duration of an interburst interval. Altogether, these results suggest a scenario in which $I_{NaP}$ initiates the burst and $I_m$ contributes to the normal oscillatory activity of pacemakers by counteracting $I_{NaP}$ during the burst.

The above predictions were explicitly tested by means of intracellular recordings of pace-maker cells driven by $I_{NaP}$ and triggered by removing the $Ca^{2+}$ from extracellular solution [5, 10, 49]. In this recording condition, the burst termination did not involve $I_{KCa}$ but engaged a $K^+$ current as the broad-spectrum $K^+$ channel blocker TEA strongly increased the duration of bursts until ultimately, a plateau-like depolarization developed ($P < 0.05$, S7C and S7D Fig). This result led us to consider the involvement of a noninactivating TEA-sensitive $K^+$ current such as $I_M$ [44]. In line with computational predictions, the blockade of $I_M$ by XE991 increased the duration and decreased the frequency of bursts ($P < 0.05$, Fig 7E and 7G). Similar results were obtained with linopirdine (S7E and S7F Fig). The ability of ICA73 to achieve the converse through the magnification of $I_M$ was found ($P < 0.05$, Fig 7F and 7G). The effects of ICA73 on burst dynamics were reproduced with retigabine (S7G and S7H Fig). Interestingly, as pre-dicted by the model, the enhancement of $I_M$ by ICA73 converted rhythmic bursting into tonic spiking in a few cells (S7I and S7J Fig). Finally, to investigate the contribution of Kv7.2 chan-nels, pacemaker cells were recorded from Kv7.2$^{Thr274Met/+}$ mice in $[Ca^{2+}]_o$-free saline. Pace-maker properties displayed long-lasting and low-frequency bursts relative to those recorded in Kv7.2$^{+/+}$ littermates ($P < 0.05$, $P < 0.01$, Fig 7H and 7I).

Together, these data show that $I$m, at least mediated by Kv7.2-containing channels, takes a significant part in the timing/intensity control of bursts to regulate dynamics of oscillatory properties in pacemakers.

## M-current mediated by Kv7.2-containing channels controls the locomotor cycle

To investigate the role of $I_M$ in networkwide rhythmogenesis, we examined its role in the oper-ation of the locomotor rhythm-generating network by using in vitro spinal cord preparations from neonatal rats. Rostral lumbar segments (L1–L2) have a more powerful rhythmogenic capacity than the caudal ones in neonatal rats [1, 2]. A Vaseline barrier was built at the L2–L3 level to selectively superfuse the 2 compartments with different drug cocktails (Fig 8A). During bath application of N-methyl-DL aspartate (NMA)/5-hydroxytryptamine (5-HT) on both sides of the barrier to induce locomotor-like activities, the addition of XE991 (10 μM) in the highly rhythmogenic L1–L2 compartment to decrease $I_M$ led to an augmentation of the loco-motor cycle period, burst duration, and burst amplitude ($P < 0.05$, Fig 8B). Also, when prein-cubated for 30 min before the application of NMA/5-HT, XE991 strongly decreased the latency for the emergence of fictive locomotion ($P < 0.05$, S8 Fig). By contrast, augmenting $I_m$ with ICA73 sped up locomotor cycles and shortened burst duration without apparent effect on burst amplitude ($P < 0.05$, Fig 8C).

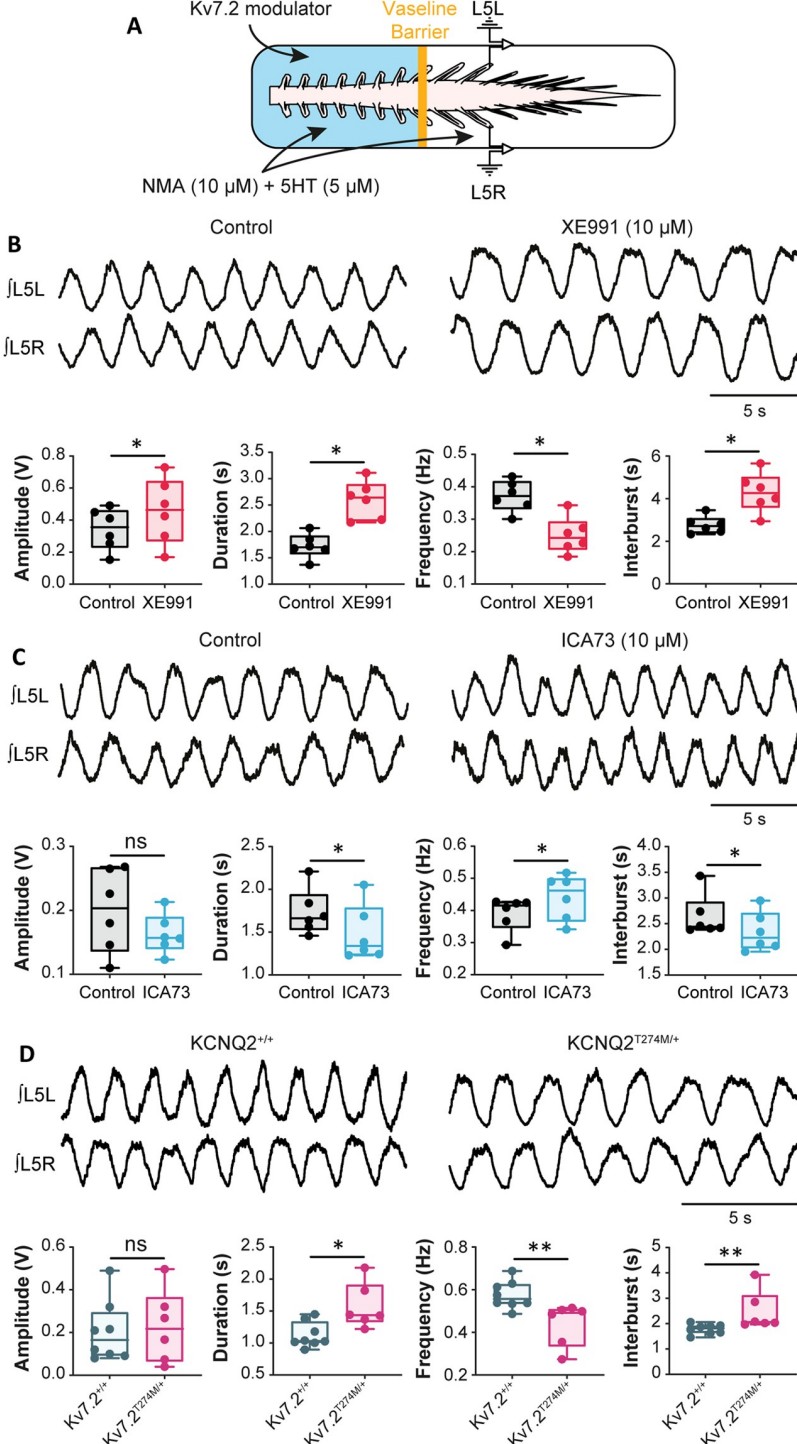

**Fig 8. $I_M$ mediated by Kv7.2-containing channels modulates the locomotor cycle at the level of the CPG.** (A) Experimental setup. (B,C) Ventral-root recordings of NMA/5-HT–induced rhythmic activity generated before and after adding XE991 (10 μM, $n$ = 5 spinal cords) (B) or ICA73 (10 μM, $n$ = 6 spinal cords) (C) to rostral lumbar segments. Below traces, boxplot quantification of locomotor burst parameters. $^*P < 0.05$, comparing data before and after the abovementioned drugs; Wilcoxon paired test. (D) Ventral-root recordings of NMA/5-HT–induced rhythmic activity generated in wild-type and Kv7.2$^{Thr274Met/+}$ mice. Below traces, boxplot quantification of locomotor burst parameters. $^*P < 0.05$, $^{**}P < 0.01$, comparing wild-type versus mutant; Mann–Whitney test. Underlying numerical values can be found in the S1 Data. CPG, central pattern generator; ICA73, $N$-(2-chloro-5-pyrimidinyl)-

3,4-diflurobenzamide; NMA, N-methyl-DL aspartate; ns, not significant; XE991, 10,10-bis(4-pyridinylmethyl)-9 (10H)-anthracenone dihydrochloride; 5-HT, 5-hydroxytryptamine.

In some CNS neurons, M-channels have been reported to increase the glutamatergic release by acting at a presynaptic level [50, 51]. Because ionotropic glutamate receptors tune the locomotor network to perform at different speeds [52], we tested the possibility that XE991 and ICA93 affect central glutamatergic synaptic transmission at the CPG level. We analyzed miniature excitatory postsynaptic currents (mEPSCs) recorded from L1–L2 ventromedial interneurons in slice preparations ($P > 0.05$, S9 Fig). Neither the amplitudes nor the frequencies of mEPSPCs were affected by Kv7 modulators.

To study the contribution of Kv7.2 channels, fictive locomotion was also induced in neonatal Kv7.2$^{\text{Thr274Met/+}}$ mice. It appeared slower ($P < 0.05$), with longer burst duration ($P < 0.01$), compared with that induced in wild-type spinal cords (Fig 8D). However, when mice acquired a full-weight–bearing stepping, juvenile Kv7.2$^{\text{Thr274Met/+}}$ mutant displayed quite normal locomotor performances (S10 Fig), suggesting a compensatory mechanism during development.

## Discussion

We provide evidence of the expression of $I_M$ in locomotor-related interneurons such as Hb9 cells and of its critical importance for the operation of the rhythm-generating mechanisms. In sum, $I_M$ appears mainly mediated by Kv7.2-containing channels and acts in opposition to $I_{NaP}$ by modulating both the emergence and frequency regime of pacemaker cells and thereby regulates the locomotor rhythm.

$I_M$ has been previously characterized in sensory and motor neurons of the spinal cord [28, 29, 53, 54] with a significant role in dampening their excitability [27, 29] to prevent both nociception and myokymia [55, 56]. Here, we attempted to identify $I_M$ at the core of the spinal rhythm-generating network for locomotion by recording L1–L2 ventromedial interneurons and Hb9$^+$ cells [40, 41]. All of them display a persistent potassium current that shares many characteristics with $I_M$, including voltage dependence, kinetics, and pharmacology [43, 57–59]. It has been suggested that native $I_M$ can be mediated by heteromeric assemblies of Kv7.2 and Kv7.3 subunits [43, 60] with an obligatory role of Kv7.2 channels [61]. A coexpression of both Kv7 subunits occurs in spinal dorsal root ganglia and motoneurons [30, 53]. On the other hand, our immunocytochemistry showed that most neurons within the CPG region are endowed with Kv7.2 subunits but weakly expressed Kv7.3 subunits. Although we cannot exclude the possibility of some contribution by Kv7.3 channels to $I_M$, the prevalent contribution by Kv7.2 channels is most likely. This is supported by the decrease of $I_M$ in CPG cells recorded from KCNQ2$^{\text{T274M/+}}$ mutant mice and the high sensitivity of $I_M$ to low concentration of TEA. The data could be interpreted by assuming that most of the $I_M$ in CPG neurons of neonatal rodents could be carried by homomeric Kv7.2 channels. However, it is worth considering that a developmental expression of Kv7.3 leading to a progressive switch towards heteromeric Kv7.2/7.3 in adulthood might occur [45, 62]. This developmental aspect may account for the compensatory mechanism observed in juvenile KCNQ2$^{\text{T274M/+}}$ mutant mice, which display locomotor movements similar to wild-type animals. In line with the existence of a compensatory mechanism, Kv7.2$^{\text{Thr274Met/+}}$ mutant mice displayed early epileptic seizures that were generally not observed when animals grew [42].

In our model, the omission of $I_M$ depolarized the resting membrane potential, predicting a contribution of $I_M$ in setting this parameter. Such a role, previously described for $I_M$ in several neuronal systems [63–68], assumes a steady-state activation of the outward current at rest. This is in agreement with the threshold activation of $I_M$, which was found to be slightly more

negative than the resting membrane potential in ventromedial and Hb9 interneurons. On the other hand, the pharmacological inhibition of $I_M$ did not affect its resting membrane potential. This discrepancy likely results from the inefficiency of Kv7 inhibitors to block $I_M$ at membrane potentials close to the resting potential [47, 48]. In line with this assumption, interneurons from Kv7.2$^{Thr274Met/+}$ mutant mice displayed a more depolarized resting membrane potential. The $I_M$ did not appreciably alter the threshold or waveform of action potentials, probably because of the slow activation kinetics of Kv7 channels [43]. However, $I_M$ dampens neuronal excitability of CPG cells by tuning the threshold current required to fire an action potential. In fact, as originally described in frog sympathetic neurons [39], $I_M$ impacted neuronal excitability of CPG cells primarily by hyperpolarizing their membrane and reducing their input resistance. In sum, $I_M$ appears to be an important player in controlling both the excitability and the firing frequency at which CPG cells are able to fire.

The present study supports a central role for $I_M$ in shaping the firing pattern of CPG interneurons. In response to a sustained depolarization, a decrease of $I_M$ converts a spiking pattern into a bursting mode in a subset of ventromedial spinal interneurons. Within both cortical and hippocampal pyramidal neurons in mammals, the prevalence of bursting also increases after a pharmacological [51, 64, 66, 67, 69] or a genetic alteration of $I_M$ [61]. Therefore, a basal Kv7.2-containing channel activity acts as a "brake" in controlling the bursting behavior of CPG cells. This assumption accords with indirect evidence suggesting that the inhibition of an M-like current might regulate pacemaker bursting cells in spinal cord primary cultures [70].

Because $I_{NaP}$ functions as the primary mechanism for oscillatory burst generation in CPG interneurons [3, 5, 9, 10], it is conceivable that $I_M$ interacts with $I_{NaP}$ to orchestrate bursting behavior. Several pieces of evidence support the existence of a dynamic interplay between $I_M$ and $I_{NaP}$. First, our voltage-clamp recordings showed a facilitation of $I_{NaP}$ once $I_M$ is reduced. Thus, even if the activation of Kv7 channels is too slow to influence the transient sodium current associated with the spike generation (see above), $I_M$ appears fast enough to interact with $I_{NaP}$. Second, according to our model that incorporates a heterogeneous distribution of $I_{NaP}$ and $I_M$, the principal distinguishing property between bursting pacemaker versus nonpacemaker behaviors was the relative magnitude of $I_{NaP}$ to $I_M$; that is, bursting cells displayed a higher $I_{NaP}/I_M$ ratio compared with nonbursting cells. Third, a coexpression between sodium and Kv7.2 channels was found in interneurons from the CPG region at the AIS, supposed to be the primary source for both $I_M$ and $I_{NaP}$ [66, 67, 71, 72]. Altogether, our data indicate that $I_{NaP}$ and $I_M$ are ubiquitously expressed in CPG neurons and that the core biophysical mechanism for oscillatory activities relies on the spatial and temporal dynamic interactions between the 2 conductances.

In addition to regulating the prevalence of bursting cells in the CPG, $I_M$ predominantly influences bursting dynamics of pacemaker neurons. At the cellular level, the decrease of $I_M$ delays the burst termination, suggesting that $I_M$ helps to swing the membrane potential down. Another notable consequence was the increase in the interburst interval even though $I_M$ appears negligible during this period. This paradox implies that $I_M$ indirectly dictates the interburst interval and thus the rhythmic frequency of walking. The Na/K pump current has been linked to the rhythmogenic mechanism of the spinal locomotor CPG networks [22, 23, 73, 74]. Specifically, the activation of Na/K pumps in response to intracellular Na$^+$ accumulation during locomotion decelerates the frequency of locomotor rhythm by mediating a postburst hyperpolarization [23]. Because the decrease of $I_M$ increases the duration of Na$^+$($I_{Nap}$)-mediated bursts, we posit that in response to Na$^+$ accumulation, the subsequent postburst hyperpolarization mediated by the Na/K pump will be higher and therefore enable neuronal bursting and rhythmic frequency of walking at low frequencies. In sum, the indirect link between $I_M$

and the Na/K pump through the $Na^+$ accumulation during locomotor bursts might finely tune the frequency of the rhythm by regulating the interburst duration.

Our data support that $I_M$ is part of the burst-firing–activated outward current contributing to the burst termination. Rather than being exclusively controlled by $I_M$, the burst termination relies on complementary factors acting in concert. Consistent with this, the substantial increase in burst duration induced by TEA until ultimately reaching a depolarization block assumes the contribution of voltage-gated $K^+$ currents other than $I_M$ in the burst termination process. This observation is consistent with a previous report identifying at least 3 $K^+$ conductances ($I_A$, $I_{Kdr}$, and $I_{KCa}$) controlling the activity of the bursting in cultured spinal neurons [75]. Note that in our recording conditions performed in $[Ca^{2+}]_o$-free saline, no obvious role could be attributed to $I_{KCa}$. On the other hand, because bursting activities rely on $I_{NaP}$, outward currents coupled to $Na^+$ accumulation such as the $Na^+/K^+$ pump current and $Na^+$-dependent $K^+$ current might be candidates for assisting burst termination. These 2 conductances have been shown to contribute to terminating inspiratory bursts in the context of respiratory rhythm generation [76]. In conclusion, we suggest that the burst repolarization in CPG cells is related to activation of multiple $K^+$ currents including $I_M$.

Rhythmic motor systems are characterized by the ability to regulate the cycle frequency of the rhythm. Multiple outward currents contribute to modulating the locomotor rhythm [15, 16, 19, 21], but the involvement of $I_M$ in the locomotor function has never been investigated. The specific modulation of the fictive locomotor rhythm following the selective application of Kv7 modulators over the CPG demonstrates that $I_M$ represents a new ionic component by which the speed of locomotion can be tuned. The effects of $I_M$ on the controllability of the locomotor rhythm in vivo lead us to conclude that $I_M$ provides a mechanism to adapt speed of movements as circumstances demand. This investigation is the first direct evidence for the concept that $I_M$ plays a key role in controlling rhythmogenesis in the spinal locomotor network. $I_M$ is a well-established target for a range of modulators [77–80] and thus may offer a powerful means to regulate the rhythmicity of the spinal locomotor network. Regarding the initial characterization of $I_M$ through its suppression by muscarinic receptor activation [39, 81], one obvious candidate is acetylcholine. Because the inhibition of $I_M$ slowed down the locomotor rhythm, the cholinergic inhibition of $I_M$ at the CPG level might play a key role in the dynamic reconfiguration of the locomotor network by changing the relative number of bursting pacemaker cells. [82]. In line with this notion, 1) the mammalian spinal cord contains several types of cholinergic neurons [83], 2) facilitation of the endogenous cholinergic system is capable of producing fictive locomotor activity in a slow speed range with a significant contribution of muscarinic receptors [84–87], 3) cholinergic cells mainly located in laminae VII and near the central canal are recruited during locomotion [88–91], and 4) a significant proportion of CPG neurons are responsive to acetylcholine in the form of intrinsic membrane potential oscillations [86, 92]. Altogether, the neuromodulation of $I_M$ through muscarinic receptors may account, at least in part, for the cholinergic locomotor rhythm control. Monoamines such as serotonin also shape spinal motor patterns in mammals particularly by lengthening their locomotor rhythm [93]. Considering the ability of the monaminergic system to interact with Kv7 channels [94] and of serotonin to promote burst firing through a decrease of $I_M$ [95], it is conceivable that neuromodulation of $I_M$ by monoamines dynamically reconfigures the firing pattern of locomotor CPG interneurons.

Aside from neuromodulation, we indicated that the propensity of a CPG neuron to burst also depends on the ionic composition of the milieu in which it is embedded [3–5, 10]. As a consequence of activity-dependent changes in extracellular calcium ($[Ca^{2+}]_o$) and potassium ($[K^+]_o$) concentrations during locomotion, a large number of CPG interneurons are converted from regular spiking into bursting through a concomitant up-regulation of $I_{NaP}$ and reduction

of $K^+$ currents [10]. Thus, by modulating $I_{NaP}$ and $I_M$, the respective changes in $[Ca^{2+}]_o$ and $[K^+]_o$ may represent a fast and powerful mechanism to regulate bursting pacemaker cells and thereby the operation of the locomotor CPG.

Overall, this study provides new, to our knowledge, insights into the operation of the locomotor network whereby $I_M$ and $I_{NaP}$ represent a functional set of subthreshold currents that endow the locomotor CPG with rhythmogenic properties, with a behavioral role of $I_M$ in controlling the speed of locomotion.

## Materials and methods

### Ethics statement

We made all efforts to minimize animal suffering and the number of animals used. All animal care and use conformed to the French regulations (Décret 2010–118) and were approved by the local ethics committee (Comité d'Ethique en Neurosciences INT-Marseille, CE Nb A1301404, authorization Nb 2018110819197361). Experiments were performed on Wistar rats, Hb9:eGFP mice, and heterozygous KCNQ2$^{T274M/+}$ mutant mice.

### Surgery and microinjections

A chronic lumbar intrathecal catheter was implanted in young adult rats using a lumbar approach. Briefly, rats received an indwelling intrathecal catheter under anesthesia for spinal drug delivery (intraperitoneal injection of ketamine at 50 mg/kg (Imalgen, MERIAL, Lyon, France) and medetomidine at 0.25 mg/kg (Domitor, Orion Pharma, Espoo, Finland). After an incision of the skin and muscles facing the vertebrae from T11 to L6, a laminectomy of the L3 vertebra was performed. A small needle was used to perforate the dura mater, and an intrathecal polyurethane catheter (32 Ga, C08PU-RIT1301, Phymep, Paris, France) was inserted in the rostral direction until it reached the position allowing a perfusion at the height of L1–L2 spinal cord. The catheter was secured to the superficial muscle of the back, and the external end of the catheter was tunneled subcutaneously and exited at in the dorsal neck region and plugged with a piece of steel wire. The skin was stitched back with 3–0 silk sutures, and the rats were placed in individual cages for recovery. Only animals with no evidence of neurological deficits after catheter insertion were studied. Behavioral testing occurred 7 days after intrathecal catheter implantation.

### The Kcnq2$^{Thr274Met/+}$ mouse model

The Kcnq2$^{Thr274Met/+}$ mouse model was generated by homologous recombination in embryonic stem (ES) cells using a targeting vector containing regions homologous to the genomic *Kcnq2* sequences and the p.(Thr274Met) variant, a recurrent pathogenic variant identified in several patients suffering from developmental and epileptic encephalopathy. Correctly targeted 129Sv ES cell clones were injected into C57Bl/6N blastocysts implanted in pseudopregnant females. Chimerism rate was assessed in the progeny by coat color markers comparison, and the mice were bred with 129sv Cre-deleter mice to excise the neomycin selection cassette and to generate *Kcnq2*$^{Thr274Met/+}$ mice. Genotyping was performed using genomic DNA prepared from ear punch biopsies with the Direct DNA (Tail) (Viagen Biotech, Los Angeles, CA, USA). The *Kcnq2*$^{Thr274Met/+}$ animals were maintained and studied on the 129Sv genetic background. Wild-type and heterozygous knock-in animals express the same amount of Kcnq2 transcript; both alleles are equally expressed. The characterization of the *Kcnq2*$^{Thr274Met/+}$ mouse reveals that it faithfully reproduces what is expected based on the human phenotype: no gross morphological brain alterations and no neurosensory alterations before the onset of seizures

occurring at P20, followed by a high rate of unexpected death in epilepsy and important cognitive difficulties [42].

## Assessment of locomotor behaviors

Adults and juvenile animals (P15–P21) were tested when a mature pattern of locomotion occurred [96]. The CatWalkXT (v9.1, Noldus Information Technology, Wageningen, Netherlands) was employed to measure walking performance. Each animal walked freely through a corridor on a glass walkway illuminated with beams of light from below. A successful walking trial was defined as having the animal walk at a steady speed (no stopping, rearing, or grooming), and 3–5 successful trials were collected per animal. Experimental sessions typically lasted for 5–10 min. The footprints were recorded using a camera positioned below the walkway, and footprint classification was manually corrected to ensure accurate readings. The paw print parameters were then analyzed using the CatWalk software (see data analysis).

## In vitro models

Details of the in vitro preparations have been previously described [97, 98] and are only summarized here. Experiments were performed on newborn rats or mice (1–5 days old). For the whole-spinal–cord preparation, the spinal cord was transected at T10, isolated, and transferred to the recording chamber perfused with oxygenated artificial cerebrospinal fluid (aCSF). For rats, the aCSF was composed of 120 mM NaCl, 4 mM KCl, 1.25 mM $NaH_2PO_4$, 1.3 mM $MgSO_4$, 1.2 mM $CaCl_2$, 25 mM $NaHCO_3$, 20 mM D-glucose (pH 7.4; 25˚C–26˚C). For mice, the aCSF was composed of 128 mM NaCl, 4 mM KCl, 0.5 mM $NaH_2PO_4$, 1 mM $MgSO_4$, 1.5 mM $CaCl_2$, 21 mM $NaHCO_3$, 30 mM D-Glucose (pH 7.4; 25˚C–26˚C). For the slice preparation, the lumbar spinal cord was isolated in ice-cold (<4˚C) aCSF with the following composition: 232 mM sucrose, 3 mM KCl, 1.25 mM $KH_2PO_4$, 4 mM $MgSO_4$, 0.2 mM $CaCl_2$, 26 mM $NaHCO_3$, 25 mM D-glucose (pH 7.4). The lumbar spinal cord was then introduced into a 1% agar solution, quickly cooled, mounted in a vibrating microtome (Leica VT1000S; Leica Biosystems, Wetzlar, Germany), and sliced (350 μm) through lumbar segments. Slices were immediately transferred into the holding chamber filled with aCSF composed of 120 mM NaCl, 3 mM KCl, 1.25 mM $NaH_2PO_4$, 1.3 mM $MgSO_4$, 1.2 mM $CaCl_2$, 25 mM $NaHCO_3$, 20 mM D-glucose (pH 7.4; 32˚C–34˚C). Following a 1-h resting period, individual slices were transferred to a recording chamber that was continuously perfused with the same medium heated to approximately 27˚C. Slices were visualized with epifluorescence and infrared differential interference contrast (IR-DIC) microscopy using a Nikon Eclipse E600FN upright microscope (Nikon, Tokyo, Japan) coupled with a 40× water immersion lens. The image was enhanced with an infrared-sensitive CCD camera and displayed on a video monitor. The temperature regulation was provided by the CL-100 bipolar temperature controller (Warner Instruments, Holliston, MA, USA).

## In vitro recordings

For the whole-spinal–cord preparation, motor outputs were recorded from lumbar ventral roots (left/right L5) by means of glass suction electrodes connected to an AC-coupled amplifier. The ventral-root recordings were amplified (×2,000), high-pass filtered at 70 Hz, low-pass filtered at 3 kHz, and sampled at 10 kHz. Custom-built amplifiers enabled simultaneous online rectification and integration (100-ms time constant) of raw signals. Locomotor-like activity was induced by a bath application of NMA (10 μM) and 5-HT (5 μM). In some experiments, a Vaseline barrier was built at the $L_2/L_3$ level to superfuse the locomotor network located in the rostral lumbar cord independently from the more caudally located motoneurons. For the slice

preparation, whole-cell patch-clamp recordings were performed from L1–L2 interneurons using a Multiclamp 700B amplifier (Molecular Devices, San Jose, CA, USA). Interneurons located in the medial lamina VIII and adjacent to the central canal, a region proposed to contain a large part of the rhythm-generating locomotor network [2], were selected. In Hb9:eGFP transgenic mice, only GFP-positive interneurons compatible with the previously described electrophysiological profile of Hb9 interneurons—i.e., high input resistance, strong postinhibitory rebound, and absence of sag—were considered [99]. Motoneurons were visually identified as the largest cells located in layer IX. Patch electrodes (2–4 MΩ) were pulled from borosilicate glass capillaries (1.5 mm OD, 1.12 mm ID; World Precision Instruments, Sarasota, FL, USA) on a Sutter P-97 puller (Sutter Instruments Company, Novato, CA, USA) and filled with intracellular solution containing 140 mM $K^+$-gluconate, 5 mM NaCl, 2 mM $MgCl_2$, 10 mM HEPES, 0.5 mM EGTA, 2 mM ATP, 0.4 mM GTP (pH 7.3; 280–290 mOsm). Pipette and neuronal capacitive currents were canceled, and after breakthrough, the series resistance was compensated and monitored. Recordings were digitized online and filtered at 10 kHz (Digidata 1322A, Molecular Devices). The main characterization of $I_M$ was accomplished by holding the membrane potential at a relatively depolarized potential ($V_H$, −10 mV) to activate KCNQ channels and to inactivate many of the other $K^+$ channels, notably Kv1.2 channels [20]. The membrane potential was then stepped down to more hyperpolarized potentials to deactivate the KCNQ channels giving rise to slow inward current relaxation. Stepping back to −10 mV led to the reactivation of the KCNQ channels to produce slow inward relaxations. All experiments were designed to gather data within a stable period (i.e., at least 5 min after establishing whole-cell access). Action-potential–independent mEPSCs were recorded in the presence of TTX (1 μM) at a holding potential of −70 mV. NMDA and non-NMDA receptor-mediated mEPSCs were recorded with a $K^+$-gluconate–based intracellular solution (see above) and pharmacologically isolated with a combination of biccuculine (20 μM) and strychnine (1 μM) to fast GABAergic and glycinergic synapses, respectively.

## Stable CHO cell lines expressing Kv7.2 or Kv7.3

CHO cells (CHO-K1 CCL-61; LGC Standards, Teddington, UK) were cultured at 37˚C in a humidified atmosphere with 5% $CO_2$ with a Gibco F-12 Nutrient Mixture (Life Technologies, Carlsbad, CA, USA) supplemented with 10% FBS (Fetal Bovine Serum) and 100 units/mL antibiotics/antimycotics (Life Technologies). 100,000 cells in suspension were transfected using the Neon Transfection System (Life Technologies) with 1 μg of pcDNA3.1/Hygro plasmid encoding Kv7.2 or pcDNA3/Neo plasmid encoding Kv7.3. Stable cell lines were subsequently established using antibiotic selection and serial dilutions in 96-well plates.

## Immunostaining

For the slice preparation, spinal cords were immersion-fixed for 1 h in 0.25% PFA, washed in PBS, and cryoprotected overnight at 4˚C in 20% sucrose in PBS. Lumbar spinal cords (L1–L2) were then frozen in OCT medium (Tissue Tec, Dormagen, Germany), cryosectioned (20 μm), and processed for immunohistochemistry. Slices were then (i) rehydrated in PBS at room temperature (15 min); (ii) permeated with 1% Bovin Serum Albumin (BSA), 2% Natural Goat Serum (NGS), and 0.2% Triton x-100 (1 h); (iii) incubated overnight at 4 ˚C in the following affinity-purified rabbit Kv7.2 (PA1-929, residues 1–70; 1:1,000; Thermo Fisher Scientific, Waltham, MA, USA), Kv7.3 (APC-051, residues 668–686; 1:400; Alomone, Jerusalem, Israel)-specific polyclonal antibody, and mouse pan-$Na_v$ (residues 932–1,043, 1:1,000; Sigma-Aldrich, St Louis, MO, USA)-specific monoclonal antibody; (iv) washed in PBS (3 × 5 min); (v) incubated with fluorescent-conjugated secondary antibodies (Alexa 488- or 546-conjugated mouse- or

rabbit-specific antibodies [1:800 and 1:400; Life Technologies]) used for visualization of the mouse monoclonal or rabbit polyclonal antibodies, respectively, in a solution containing 1% BSA and 2% NGS (1.5 h); (vi) washed in PBS 3 × 5 min; and (vii) coverslipped with a gelatinous aqueous medium. In control experiments, the primary antiserum was replaced with negative control mouse or rabbit immunoglobulin fraction during the staining protocol. Sections were scanned using a laser scanning confocal microscope (Zeiss LSM700; Carl Zeiss, Oberkochen, Germany) in stacks of 0.3- to 1-μm–thick optical sections at ×63 and/or ×20 magnification, respectively, and processed with the Zen 12.0 software (Zeiss). Each optical section resulted from 2 scanning averages. Each figure corresponds to a projection image from a stack of optical sections. For CHO cells stably expressing Kv7.2 or Kv7.3, cells were fixed with 4% paraformaldehyde and blocked with permeabilization buffer (0.2% Triton-x100, 1% BSA, 2% normal goat serum in PBS) for 30 min at room temperature. CHO cells were incubated with the appropriate primary antibody diluted in permeabilized buffer for 60 min. After extensive PBS washing, CHO cells were incubated in blocking buffer (1% BSA, 2% normal goat serum in PBS) with secondary antibodies for 45 min at room temperature, washed, and mounted (fluorSave Reagent; Calbiochem, San Diego, CA, USA). The following antibody dilutions were used: Thermo Fisher Scientific rabbit anti-Kv7.2 PA1-929 (1:1,000), Alomone rabbit anti-Kv7.3 APC-051 (1:100), and Thermo Fisher Scientific donkey anti-rabbit Alexa Fluor 488 (1:500). Imaging was performed with a Zeiss Apotome.2 microscope using 25× oil immersion objective.

## Validation of Kv7.2 and Kv7.3 antibodies on CHO cells

We established the specificity of the Kv7.2 and Kv7.3 antibodies on CHO cells stably expressing Kv7.2 or Kv7.3. The CHO cells stably expressing Kv7.2 were strongly immunostained with the Kv7.2 antibody, and no labeling above background level was observed with the Kv7.3 antibody (S2 Fig). The CHO cells stably expressing Kv7.3 were immunostained with the Kv7.3 antibody, and no labeling above background level was observed with the Kv7.2 antibody (S2 Fig). In sum, these data provide solid evidence of the specificity of the antibodies against Kv7.2 and Kv7.3 channels with no cross-reactivity. Note that the specificity of the Kv7.3 antibody has been validated previously [61].

## Cell model

The model of the pacemaker neuron is a typical somatic single-compartment model developed in the Hodgkin–Huxley style and was based on our previous study on Hb9 cells [10]. The neuronal membrane potential V was dynamically defined by a set of membrane ionic currents. The current balance equation is

$$C \cdot \frac{dV}{dt} \ = \ -I_{\text{Na}} - I_{\text{NaP}} - I_{\text{K}} - I_{\text{M}} - I_{\text{Pump}} - I_{\text{L}} + I_{\text{Inj}},$$

where $C$ is neuronal membrane capacitance (pF) and $t$ is time (ms). The modeled neuron included the following ionic currents: transient sodium current ($I_{\text{Na}}$ with the maximal conductance $\bar{g}_{\text{Na}}$), the persistent sodium current ($I_{\text{NaP}}$ with the maximal conductance $\bar{g}_{\text{NaP}}$), the noninactivating M-current ($I_{\text{M}}$ with the maximal conductance $\bar{g}_{\text{M}}$), the delayed rectifier potassium current ($I_{\text{K}}$ with the maximal conductance $\bar{g}_{\text{K}}$), the sodium-dependent pump ($I_{\text{Pump}}$), leakage ($I_{\text{L}}$ with the conductance $g_{\text{L}}$), and depolarizing injected ($I_{\text{Inj}}$) currents. These currents, except for $I_{\text{Pump}}$ and $I_{\text{Inj}}$, are described as follows:

$$I_{\text{Na}} \ = \ \bar{g}_{\text{Na}} \cdot m_{\text{Na}}^3 \cdot h_{\text{Na}} \cdot (V - E_{\text{Na}});$$

$$I_{\mathrm{NaP}} = \bar{g}_{\mathrm{NaP}} \cdot m_{\mathrm{NaP}} \cdot h_{\mathrm{NaP}} \cdot (V - E_{\mathrm{Na}});$$

$$I_{\mathrm{K}} = \bar{g}_{\mathrm{K}} \cdot m_{\mathrm{K}}^{4} \cdot (V - E_{\mathrm{K}});$$

$$I_{\mathrm{M}} = \bar{g}_{\mathrm{M}} \cdot m_{\mathrm{M}} \cdot (V - E_{\mathrm{K}});$$

$$I_{\mathrm{L}} = g_{\mathrm{L}} \cdot (V - E_{\mathrm{L}}).$$

$E_{\mathrm{Na}}$, $E_{\mathrm{K}}$, and $E_{\mathrm{L}}$ are reversal potentials of the corresponding channels (in mV):

$$E_{\mathrm{Na}} = 26.54 \cdot \ln([\mathrm{Na}^+]_\mathrm{o}/[\mathrm{Na}^+]_\mathrm{i});$$

$$E_{\mathrm{K}} = 26.54 \cdot \ln([\mathrm{K}^+]_\mathrm{o}/[\mathrm{K}^+]_\mathrm{i});$$

$$E_{\mathrm{L}} = 26.54 \cdot \ln\left(\frac{[\mathrm{K}^+]_\mathrm{o} + p_{\mathrm{Na/K}} \cdot [\mathrm{Na}^+]_\mathrm{o} + p_{\mathrm{Cl/K}} \cdot [\mathrm{Cl}^-]_\mathrm{i}}{[\mathrm{K}^+]_\mathrm{i} + p_{\mathrm{Na/K}} \cdot [\mathrm{Na}^+]_\mathrm{i} + p_{\mathrm{Cl/K}} \cdot [\mathrm{Cl}^-]_\mathrm{o}}\right).$$

[Na$^+$], [K$^+$], and [Cl$^-$] represent concentrations of sodium, potassium, and chloride ions, respectively. Indexes "o" and "i" define the concentrations of these ions outside and inside a cell, respectively: $[\mathrm{Na}_+]_\mathrm{o}$ = 145 mM, $[\mathrm{K}_+]_\mathrm{o}$ = 3 mM, $[\mathrm{K}_+]_\mathrm{i}$ = 140 mM, $[\mathrm{Cl}^-]_\mathrm{i}$ = 8 mM, $[\mathrm{Cl}^-]_0$ = 130 mM, and $[\mathrm{Na}_+]_\mathrm{i}$ is considered variable. Parameters $p_{\mathrm{Na/K}}$ and $p_{\mathrm{Cl/K}}$ represent the relative permeability of sodium and chloride ions (with respect to potassium ions): $p_{\mathrm{Na/K}}$ = 0.03, $p_{\mathrm{Cl/K}}$ = 0.1.

In our model, the intracellular sodium concentration $[\mathrm{Na}^+]_\mathrm{i}$ is accumulating because of $I_{\mathrm{Na}}$ and $I_{\mathrm{NaP}}$ currents and is pumped out by $I_{\mathrm{Pump}}$. The contributions of these components are defined by the corresponding coefficients ($\alpha_{\mathrm{Na}}$, $\alpha_{\mathrm{NaP}}$, and $\alpha_{\mathrm{Pump}}$, respectively):

$$\frac{d}{dt}[\mathrm{Na}^+]_\mathrm{i} = -\alpha_{\mathrm{Na}} \cdot I_{\mathrm{Na}} - \alpha_{\mathrm{NaP}} \cdot I_{\mathrm{NaP}} - \alpha_{\mathrm{Pump}} \cdot 3 \cdot I_{\mathrm{Pump}}$$

The pump current is described as $I_{\mathrm{Pump}} = R_{\mathrm{Pump}} \cdot (\phi([\mathrm{Na}^+]_\mathrm{i}) - \phi([\mathrm{Na}^+]_{\mathrm{ibase}})$, where $\phi(y) = y^3/(y^3 + K_{\mathrm{Pump}}^3)$, $[\mathrm{Na}^+]_{\mathrm{ibase}}$ is the base intracellular sodium concentration, and $R_{\mathrm{Pump}}$ and $K_{\mathrm{Pump}}$ are the $I_{\mathrm{Pump}}$ parameters. The following parameters were used for sodium dynamics and pump current: $\alpha_{\mathrm{Na}} = \alpha_{\mathrm{NaP}} = \alpha_{\mathrm{Pump}} = 10^{-5}$ mM/fC, $R_{\mathrm{Pump}}$ = 60 pA, $[\mathrm{Na}+]_{\mathrm{ibase}}$ = 15 mM, and $K_{\mathrm{Pump}}$ = 18 mM.

The dynamics of the activation ($m$) or inactivation ($h$) variables for the above sodium and potassium channels are generally described by the differential equation $dx/dt = (x_\infty - x)/\tau_x$, $x = \{m, h\}$, where $x_\infty$ is the voltage-dependent steady-state value and $\tau_x$ is the voltage-dependent time constant of the variable $x$, which are described in the following form:

$$x_\infty = 1/(1 + exp(-(V - V_{x1/2})/k_x));$$

$$\tau_x = \tau_{xmax}/\cosh\left(\frac{-(V - V_{x1/2})}{k_{\tau_{xmax}}}\right).$$

$V_{x1/2}$ and $k_x$ are the half-activation voltage and the slope for variable $x$, $\tau_{xmax}$ is the maximum value of its time constant, and $k_\tau$ defines the slope of this time constant. The activation of sodium currents ($I_{\mathrm{Na}}$ and $I_{\mathrm{NaP}}$) is considered instant, i.e., $\tau_{m\mathrm{Na}} = \tau_{m\mathrm{NaP}} = 0$; thus, $m_{\mathrm{Na}}$ and $m_{\mathrm{NaP}}$ are considered equal to their steady-state values.

**Table 2.**

| Current | Parameters |
|---|---|
| Fast Na$^+$ | $g_{Na}$ = 110 ± 11 nS |
| | *Act*: V$_{mNa1/2}$ = −43.8 mV; $k_{mNa}$ = 6 mV |
| | *Inact*: V$_{hNa1/2}$ = −67.5 mV; $k_{hNa}$ = −10.8 mV; $\tau_{hNamax}$ = 35.2 ms; $k_{rhNa}$ = 12.8 mV |
| Persistent Na$^+$ | $g_{NaP}$ = 0.4 ± 0.1 nS |
| | *Act*: V$_{mNaP1/2base}$ = −52 ± 1.5 mV; $k_{mNaP}$ = 3 mV |
| | *Inact*: V$_{hNaP1/2}$ = −59 mV; $k_{hNaP}$ = −5.2 mV; $\tau_{hNaPmax}$ = 20,000 ms; $k_{rhNaP}$ = 10.4 mV |
| K$^+$ delayed rectifier | $g_K$ = 60 ± 6 nS |
| | *Act*: V$_{mK1/2}$ = −34.5 mV; $k_{mK}$ = 5 mV; $\tau_{mKmax}$ = 4 ms; $k_{rmK}$ = 30 mV |
| M-current | $g_M$ = 0.8 ± 0.1 nS |
| | Act: V$_{mM1/2}$ = −44 ± 1.5 mV; $k_{mM}$ = 4.3 mV; $\tau_{mMmax}$ = 7 ms; $k_{rmM}$ = 25 mV |
| Other | C = 21.5 pF; $g_L$ = 1 ± 0.1 nS |

To reproduce our experimental finding, the half-activation voltage for $I_{NaP}$, V$_{mNaP1/2}$, is made dependent on the outside calcium concentration:

$$V_{mNaP_{1/2}} = V_{mNaP_{1/2}base} - \Delta V_{Ca} \cdot ([Ca^{2+}]_{obase} - [Ca^{2+}]_o),$$

where V$_{mNaP1/2base}$ corresponds to the half-activation voltage at $[Ca^{2+}]_o = [Ca^{2+}]_{obase}$ = 1.2 mM and $\Delta V_{Ca}$ = 1 mV/mM defines a shift in V$_{mNaP1/2}$ occurring with changes in $[Ca^{2+}]_o$.

In our simulations, we considered either a single neuron or a population of 50 uncoupled neurons. To provide a necessary heterogeneity in properties of neurons, we Gaussian-distributed the base values V$_{1/2}$ for $I_M$ and $I_{NaP}$ derived from our recordings. An additional heterogeneity was set by normal distribution of all conductances around appropriate base values (Table 2). Parameters for $I_{Na}$, $I_K$, and inactivation of $I_{NaP}$ were taken from previous modeling studies with some modifications.

Simulations were performed using MATLAB R2019b (The MathWorks, Natick, MA, USA). Differential equations were solved using a variable-order multistep differential equation solver ode15s available in MATLAB. In each simulation, a settling period of 30 s was allowed before data were collected. For population, each simulation was repeated 10 times, and demonstrated qualitatively similar behavior for particular values of parameters within the standard deviations.

## Data analysis

**Quantitative gait analyses.** The CatWalk XT software (v9.1, Noldus Information Technology) was used to measure a broad number of spatial and temporal gait parameters in several categories. These include i) dynamic parameters related to individual paw prints, such as duration of the step cycle with the respective duration of the swing and stance phases; ii) parameters related to the position of paw prints with respect to each other, for example, the stride length (distance between 2 consecutive placements of the same paw) and the base of support (the width between both the front and hind paws); and iii) parameters related to time-based relationships between paw pairs, as well as step patterns. These parameters were initially calculated for each run and for each paw, then averaged over the runs, and finally, the values for the front and hind paws were averaged. To determine which parameters were affected, data were normalized against basal values, which were settled as 100% in each parameter.

**Electrophysiological data analyses.** Clampfit 10.7 software (Molecular Devices) was used for analyzing electrophysiological data. Alternating activity between right/left L5 recordings

was taken to be indicative of fictive locomotion. To characterize locomotor burst parameters, raw extracellular recordings from ventral roots were rectified, integrated, and resampled. Peak amplitude of locomotor burst was measured, and the cycle period was calculated by measuring the time between the first 2 peaks of the autocorrelogram. The coupling between right/left L5 was estimated by measuring the correlation coefficient of the cross-correlogram at zero phase lag. The onset of a locomotor-like activity was determined when a clear rhythmic alternating activity was observed. Several basic criteria were set to ensure optimum quality of intracellular recordings. Only cells exhibiting a stable resting, holding membrane potential, access resistance (less than 20% variation), and an action potential amplitude larger than 40 mV were considered. All reported membrane potentials were corrected for liquid junction potentials. We determined input resistance by the slope of linear fits to small ($<5$ mV) voltage responses evoked by positive and negative current injections. Firing properties were measured from depolarizing current pulses of varying amplitudes. The rheobase was defined as the minimum step current intensity required to induce an action potential from the membrane potential held at −60 mV. Single-spike analysis was performed on the first spike elicited near the rheobase. Peak spike amplitude was measured from the threshold potential, and spike duration was defined as the time to fall to half-maximum peak. The instantaneous discharge frequency was determined as the inverse of interspike interval and plotted as a function of time. For a direct comparison of firing properties before and during the application of the drug, a bias current could be used to maintain the membrane potential at the holding potential fixed in the control condition. Voltage dependence and kinetics of $I_M$ were analyzed from data normalized to the maximal current. The $I–V$ curves were fitted with a Boltzmann function. $I_M$ amplitude was measured from deactivation relaxation at -50mV. mEPSCs were detected and analyzed using the MiniAnalysis Program (Synaptosoft, Decatur, GA, USA). Events were detected by setting the threshold value for detection at 3 times the level of the root mean-square noise (approximately 3–4 pA, meaning detection threshold at approximately 8–12 pA). The average values of mPSCs amplitude and frequency during the control period and 30 min after the drug application, were calculated over a 5-min time window.

**Immunohistochemistry analysis.** Neurons with AISs of obvious soma origin were imaged. Image stacks were converted into single maximum intensity z-axis projections and imported into MATLAB for analysis using a previously published MATLAB code [100] downloaded from the Grubb Lab (ais_z3.m from http://grubblab.org/resources/). Measurements were performed on initial segments from interneurons located in the ventromedial part of upper lumbar segments (L1–L2) near the central canal and from motoneurons identified as the biggest cells located in the ventral horn. Initial segments were identified as linear structures labeled by pan-Na$_v$–specific antibodies and for which the beginning and the end of the structure could be clearly determined, excluding nodes of Ranvier. We drew a line profile starting at the soma that extended down the axon, through and past the AIS. For quantifications of the start and end position of immunolabeling along the axonal process from the soma, axonal profiles were smoothed using an approximately 5 μm sliding mean and normalized to the maximum smoothed fluorescence. AIS start and end positions were identified as the points at which fluorescence intensities increased above and dropped below 33% of the maximum axonal fluorescence intensity, respectively, in line with previous reports [100].

## Drug list and solutions

Normal aCSF was used in most cases for in vitro electrophysiological recordings. $Ca^{2+}$-free solution was made by removing $Ca^{2+}$ chloride from the recording solution and replacing it with an equimolar concentration of magnesium chloride. All solutions were oxygenated with

95% $O_2$/5% $CO_2$. For whole-cell voltage-clamp recordings of $I_M$, the aCSF was supplemented with 1 μM TTX. All salt compounds, TEA, kynurenic acid, strychnine, biccuculine, riluzole, NMA, and 5-HT were obtained from Sigma-Aldrich. Other drugs, including XE991 ($IC_{50}$ = 0.98 μM [43]), 1,3-dihydro-1-phenyl-3,3-*bis*(4-pyridinylmethyl)-2*H*-indol-2-one dihydrochloride (linopirdine, $IC_{50}$ = 4.8 μM [43]), ICA-069673 or ICA73 ($EC_{50}$ = 0.69 μM [36]), ethyl [2-amino-4-[[(4-fluorophenyl)methyl]amino]phenyl] carbamate (retigabine, $EC_{50}$ = 0.6 μM [34]), 2- amino-6-trifluoromethoxybenzothia-zole hydrochloride (riluzole), and TTX were obtained from Tocris Bioscience (Nordenstadt, Germany). Riluzole, retigabine, and ICA73 were dissolved in DMSO and added to the aCSF (final concentration of DMSO: 0.05%). The other drugs were dissolved in water and added to the aCSF. Control experiments showed no effects of the vehicle.

### Treatment design

Adult and juvenile rats (15–to 21 days old) were randomly treated with a single dose of linopirdine, retigabine, XE991, or ICA73 or its vehicle. The dose of drugs refers to previous reports [31–33]. All the drugs and their vehicle were administered intraperitoneally or intrathecally. The behavioral test was performed before drug treatment and 30 min after i.p. or 5–10 min after i.t. injection of drug or vehicle. The drugs i.t. administrated were delivery in strict respect of a final injected volume of 50 μl. The volume of ICA (0.05 mg/kg) and DMSO was 20 μl followed by 30 μl of saline, and that of XE991 was 10 μL (0.025 mg/kg) followed by 40 μl of saline.

### Statistics

No statistical method was used to predetermine sample size. Group measurements were expressed as means ± SEM. We used Mann–Whitney test or Wilcoxon matched pairs test to compare 2 groups. For all statistical analyses, the data met the assumptions of the test, and the variance between the statistically compared groups was similar. The level of significance was set at $P < 0.05$. Statistical analyses were performed using Prism 5.0 software (GraphPad). Boxplots in figures show the distribution extremes (horizontal bars), 25th and 75th percentiles (box height), and median (center bar).

### Supporting information

**S1 Fig. $I_M$ does not affect interlimb coordination and gait.** (A–H) Normalized changes of the stride length (A,E), the base of support (B,F), the number of normal step sequence patterns (C,G), and regularity index of paw placements (D,H) during CatWalk locomotion of juvenile rats (A–D) before and 30 min after acute i.p. administration of DMSO (gray, $n$ = 6 rats), retigabine (5 mg/kg, green, $n$ = 7 rats), linopirdine (3 mg/kg, yellow, $n$ = 6 rats), ICA73 (5 mg/kg, blue, $n$ = 7 rats), or XE991 (5 mg/kg, red, $n$ = 6 rats) and of adult rats (E–H) before and 5–10 min after acute i.t. administration at the L1–L2 level of saline (purple, $n$ = 10 rats), DMSO (gray, $n$ = 5 rats), ICA73 (0.05 mg/kg, blue, $n$ = 8 rats), or XE991 (0.025 mg/kg, red, $n$ = 6 rats). Dashed lines with gray shading indicate the 95% confidence intervals of control values. ns, $P > 0.05$, comparing data collected before and after drug administration; Wilcoxon paired test. Underlying numerical values can be found in the S1 Data. ICA73, *N*-(2-chloro-5-pyrimidinyl)-3,4-difluorobenzamide; i.p., intraperitoneal; i.t., intrathecal; ns, not significant; XE991, 10,10-bis(4-pyridinylmethyl)-9(10H)-anthracenone dihydrochloride.
(TIF)

**S2 Fig. Lumbar motoneurons express Kv7.2-containing channels.** (A–F) Immunostaining of lumbar (L1–L2) motoneurons from juvenile rats ($n$ = 3 rats) against Kv7.2 (A, $n$ = 85 cells)

or Kv7.3 (D, *n* = 104 cells) along the AIS labeled by the pan-Na$_v$ antibody (B,E). Kv7.2 and pan-Na$_v$ are merged in (C), and Kv7.3 and pan-Na$_v$ are merged in (F). Asterisks indicate the nucleus position and arrowheads the AIS. Scale bars = 20 μm. (G) Group means quantification of the proportion of pan-Na$_v$ positive motoneurons expressing Kv7.2 or Kv7.3 channels. (H) Group means quantification of the start and end positions of pan-Na$_v$, Kv7.2, and Kv7.3 immunolabeling along the axonal process from the soma (*n* = 10 cells). $^*P < 0.05$, $^{**}P < 0.01$, comparing start or end positions between groups; Mann–Whitney test. Data are mean ± SEM. Underlying numerical values can be found in the S1 Data. AIS, axonal initial segment; Na$_v$, voltage-gated sodium channel.
(TIF)

**S3 Fig. Antibodies against Kv7.2 or Kv7.3 are specific and do not cross-react.** Immunostaining of Kv7.2-expressing (*top*) or Kv7.3-expressing (*bottom*) CHO cells in the presence of the Kv7.2 (*left*) or Kv7.3 (*right*) antibody. Scale bar: 50 μm. CHO, Chinese hamster ovary.
(TIF)

**S4 Fig. The $I_M$ enhancer retigabine replicates the effect of ICA73 on electroresponsive properties of L1–L2 ventromedial interneurons.** (A) Representative deactivation of $I_M$. (B) Boltzmann-fitted *I–V* relationships of $I_M$. (C–F) Boxplot quantification of the amplitude (C), holding current (D), threshold (E), and $V_{1/2}$ (F) of $I_M$ recorded in interneurons (*n* = 7 cells) before (black) and after bath-applying retigabine (100 nM, green). $^*P < 0.05$; Wilcoxon paired test. (G, H) Typical spiking activity of ventromedial interneurons (L1–L2) to a near-threshold depolarizing pulse (G) with the respective frequency–current relationship (H) before (black) and after bath-applying retigabine (100 nM, green). The continuous line is the best-fitting linear regression. $^{***}P < 0.001$, comparison of the fits. Underlying numerical values can be found in the S1 Data. ICA73, *N*-(2-chloro-5-pyrimidinyl)-3,4-difluorobenzamide; *I–V*, current–voltage.
(TIF)

**S5 Fig. Characterization of $I_M$ in Hb9 locomotor-related interneurons.** (A–C) Representative deactivation (A), Boltzmann-fitted *I–V* relationships (B), and amplitude (C) of $I_M$ recorded in Hb9$^+$ interneurons from Hb9:eGFP transgenic neonatal mice under control conditions (*n* = 8 cells, black) or in the presence of XE991 (10 μM, *n* = 6 cells, red). $^*P < 0.05$; Mann–Whitney test. Data are mean ± SEM. Underlying numerical values can be found in the S1 Data. eGFP, enhanced green fluorescent protein; *I–V*, current–voltage; XE991, 10,10-bis (4-pyridinylmethyl)-9(10H)-anthracenone dihydrochloride.
(TIF)

**S6 Fig. The $I_M$-blocker XE991 antagonizes the effect of ICA73 on electroresponsive properties of L1–L2 ventromedial interneurons.** (A) Membrane potential changes in response to bath application of ICA73 (10 μM) before (light blue) and after (dark blue) pretreatment with XE991 (10 μM). (B,C) Representative spiking activity to a near-threshold depolarizing step (B) and frequency–current relationship (C) recorded under XE991 before (red) and after bath-applying ICA73 (*n* = 5 cells, dark blue). Continuous lines are the best-fitting linear regression. ns, $P > 0.05$ comparison of the fits. (D) Boxplot quantification of the rheobase. ns, $P > 0.05$; Wilcoxon paired test. Underlying numerical values can be found in S1 Data. ICA73, *N*-(2-chloro-5-pyrimidinyl)-3,4-difluorobenzamide; ns, not significant; XE991, 10,10-bis(4-pyridinylmethyl)-9(10H)-anthracenone dihydrochloride.
(TIF)

**S7 Fig. $I_M$ controls burst dynamics.** (A) Firing behavior from a bursting pacemaker neuron model in response to 2 different values of $g_M$. (B) Dependence of the percentage of bursting

cells on $g_M$ in the heterogeneous population model of 50 neurons. In A and B. $V_{1/2}$ $I_{NaP}$ = −54 mV. (C,E,G,I) $[Ca^{2+}]_o$-free-saline–induced bursting activity recorded intracellularly in L1–L2 ventromedial interneurons before and after TEA (10 mM, $n$ = 7 cells) (C), linopirdine (10 μM, $n$ = 10 cells) (E), retigabine (100 nM, $n$ = 6 cells) (G), or ICA73 (10 μM, $n$ = 9 cells) (I). (D,F,H) Boxplot quantification of the duration and frequency of bursts. $^*P < 0.05$, $^{**}P < 0.01$, comparing data before and after the abovementioned drugs; Wilcoxon paired test. (J) Proportion of burster and nonburster interneurons before and after bath-applying ICA73 ($n$ = 9 cells). Underlying numerical values can be found in the S1 Data. ICA73, $N$-(2-chloro-5-pyrimidinyl)-3,4-difluorobenzamide; TEA, tetraethylammonium.
(TIF)

**S8 Fig. Down-regulation of $I_M$ facilitates the emergence of locomotor-like activity.** (A, B) Ventral-root recordings of NMA/5-HT–induced rhythmic activity generated without ($n$ = 9 spinal cords) (A) and with ($n$ = 6 spinal cords) a 45-min preincubation of XE991 (10 μM) (B). Broken-line boxes indicate the parts of the recordings that are enlarged in insets to visualize the onset of locomotor-like activity (rhythmic alternating activity). (C) Boxplot quantification of the delay between the start of the bath application of NMA/5-HT and the onset of the fictive locomotion. $^*P < 0.05$; Mann–Whitney test. Underlying numerical values can be found in the S1 Data. NMA, N-methyl-DL aspartate; XE991, 10,10-bis(4-pyridinylmethyl)-9(10H)-anthracenone dihydrochloride; 5-HT, 5-hydroxytryptamine.
(TIF)

**S9 Fig. $I_M$ has no effect on glutamatergic synaptic transmission.** (A) Representative current traces of continuously recorded mEPSCs in a ventromedial interneuron voltage clamped at −60 mV. mEPSCs recorded before and 30 min after adding XE991 (10 μM, $n$ = 6 cells) or ICA73 (10 μM, $n$ = 6 cells) were pharmacologically isolated in the presence of TTX (0.5 μM), strychnine (1 μM), and bicuculline (20 μM). (B) Averaged mEPSCs recorded before (black) and 30 min after adding XE991 (red) or ICA73 (blue). (C) Boxplot quantification of the mean frequency and amplitude of mEPSCs before and 30 min after XE991 or ICA73 was bath-applied. Dotted lines indicate the 95% confidence intervals of control values. ns, $P > 0.05$, comparing data collected before and after bath-applying the abovementioned drugs; Wilcoxon paired test. Underlying numerical values can be found in the S1 Data. ICA73, $N$-(2-chloro-5-pyrimidinyl)-3,4-difluorobenzamide; mEPSC, miniature excitatory postsynaptic current; ns, not significant; TTX, tetrodotoxin; XE991, 10,10-bis(4-pyridinylmethyl)-9(10H)-anthracenone dihydrochloride.
(TIF)

**S10 Fig. Juvenile Kv7.2$^{Thr274Met/+}$ mutant mice display normal locomotor movements.** (A) Representative footfall diagrams during CatWalk locomotion of juvenile wild-type (gray, $n$ = 7 mice) or Kv7.2$^{Thr274Met/+}$ mutant mice (pink, $n$ = 6 mice). The stance phase is indicated by horizontal bars and the swing phase by open spaces. (B) Boxplot quantification of the body speed. ns, $P > 0.05$, comparing wild-type versus Kv7.2$^{Thr274Met/+}$ mutant mice; Mann–Whitney test. (C) Quantification of swing and stance phases. ns, $P > 0.05$, comparing wild-type versus Kv7.2$^{Thr274Met/+}$ mutant mice Mann–Whitney test. Data in C are mean ± SEM. Underlying numerical values can be found in the S1 Data. LF, left forelimb; LH, left hindlimb; ns, not significant; RF, right forelimb; RH, right hindlimb.
(TIF)

**S1 Data. Raw data used in this study.**
(XLSX)

## Acknowledgments

We are grateful to the lab members for their critical reading of the manuscript and Anne Duhoux for animal care. We thank the company NSrepair headed by Philippe Marino for performing intrathecal surgery and postoperative care.

## Author Contributions

**Conceptualization:** Frédéric Brocard.

**Formal analysis:** Jérémy Verneuil, Cécile Brocard.

**Funding acquisition:** Frédéric Brocard.

**Investigation:** Jérémy Verneuil, Cécile Brocard, Laurent Villard, Julie Peyronnet-Roux, Frédéric Brocard.

**Methodology:** Virginie Trouplin.

**Resources:** Laurent Villard.

**Supervision:** Julie Peyronnet-Roux, Frédéric Brocard.

**Validation:** Frédéric Brocard.

**Writing – original draft:** Jérémy Verneuil.

**Writing – review & editing:** Frédéric Brocard.

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
