## [Editor Report · Decision Letter 0]

20 Apr 2020

Dear Dr Brocard, 

Thank you for submitting your manuscript entitled "It takes two to tango: M-current swings with the persistent sodium current to set the speed of locomotion." for consideration as a Research Article by PLOS Biology.

Your manuscript has now been evaluated by the PLOS Biology editorial staff, as well as by an Academic Editor with relevant expertise, and I am writing to let you know that we would like to send your submission out for external peer review. Please accept my apologies for the delay in communicating this decision to you, but due to COVID-19, our internal processes as well as our communication with external contributors, including Academic Editors, have been delayed. 

Please note that, before we can send your manuscript to reviewers, we need you to complete your submission by providing the metadata that is required for full assessment. To this end, please login to Editorial Manager where you will find the paper in the 'Submissions Needing Revisions' folder on your homepage. Please click 'Revise Submission' from the Action Links and complete all additional questions in the submission questionnaire.

Please re-submit your manuscript within two working days, i.e. by Apr 22 2020 11:59PM.

Kind regards,

Gabriel Gasque, Ph.D.,

Senior Editor

PLOS Biology

---

## [Decision Letter · Decision Letter 1]

1 Jun 2020

Dear Dr Brocard,

Thank you very much for submitting your manuscript "It takes two to tango: M-current swings with the persistent sodium current to set the speed of locomotion." for consideration as a Research Article at PLOS Biology. Your manuscript has been evaluated by the PLOS Biology editors, by an Academic Editor with relevant expertise, and by four independent reviewers. You will note that reviewers 1, Tuan Bui, and 2, Andrea Nistri, have signed their comments. 

In light of the reviews (below), we will not be able to accept the current version of the manuscript, but we would welcome re-submission of a much-revised version that takes into account the reviewers' comments. We cannot make any decision about publication until we have seen the revised manuscript and your response to the reviewers' comments. Your revised manuscript is also likely to be sent for further evaluation by the reviewers.

We expect to receive your revised manuscript within 2 months. 

**IMPORTANT - SUBMITTING YOUR REVISION**

Your revisions should address the specific points made by each reviewer. As you will see, the four reviewers are thoughtful and generally positive. Having discussed their comments with the Academic Editor, we think you should experimentally address the major concerns of reviewer 3. Because your article focuses on speed control, we think targeting the L1/2 segments of the spinal cord with agonists and antagonists would be very important. Although we understand the experiment might not be trivial, the Academic Editor thinks it should be doable in juvenile rats and mice (postnatal day 15 to 20) as in Figure 1. In addition, as pointed out by reviewer 4, we think that a better pharmacological characterization of the M-current would also be helpful, and gait assessments of mutant mice and intracellular recordings of mutant spinal interneurons will also be very important for addressing the legitimate issues raised by this reviewer regarding modeling experiments. 

Please submit the following files along with your revised manuscript:

*Re-submission Checklist*

*Published Peer Review*

*PLOS Data Policy*

*Blot and Gel Data Policy*

Sincerely,

Gabriel Gasque, Ph.D., 

Senior Editor

PLOS Biology

REVIEWS:

Reviewer #1, Tuan Bui: The spinal cord has the remarkable ability to generate organized, sustained rhythmic activity underlying locomotor activity. However, the operation of spinal locomotor networks is not fully understood. This study uses a combination of behavioural testing, immunohistochemistry, pharmacology, computational modelling, and electrophysiology in a number of spinal neurons and spinal cord preparations to show that the subthreshold non-inactivating potassium M-current acts as a break to counterbalance the persistent sodium current that has been previously shown to be one of the principal rhythmogenic drive of spinal locomotor networks. Using both enhancers and blockers of Kv7 channels or their sub-units as well as transgenic mice with mutations to the gene encoding Kv7.2, the study characterizes this current in spinal neurons and shows that this current terminates bursts in spinal neurons. Their computational modelling allows them to support the conclusions about the role of Kv7 channels inferred from their experimental work 

I have no major issues with the study. It provides invaluable insights that will further our understanding of the operation of spinal locomotor networks by opening lines of investigations into how this current is modulated by spinal and supraspinal pathways to regulate locomotor activity. 

Minor issues:

1. Page 9, Fig 4 data, does the model replicate the translation of the I-F curve by enhancement of the M-current without a change in slope? Or if it did not because Kv7 inhibitors are less efficient at hyperpolarized membrane potential, could a model of M-current where the M-current is blocked at more depolarized membrane potential to replicate the potential-dependance of the inhibitors show the same changes in the I-F curves? 

2. Page 10, line 3, could you estimate the percentage of sampled ventromedial interneurons where blocking the M-current enhanced a iNaP? Fig. 4 G-H suggests that this might only be found in 18.2% of the population.

3. Page 11, first paragraph, was a prolongation of the interburst by M-current blockade found in either the in-vitro locomotor activity with M-current blockade or the mutant KCNQ line?

4. Page 11, line 24, missing "of" between "blockade" and "Im"

5. Page 12, line 5 "converted rhythmic bursting to spiking into few cells". Do you mean "converted rhythmic bursting into tonic spiking in a few cells"?

6. Page 14, line 17, remove Pan. 

7. Page 28, line 10, change "depended" to "dependent.

8. Page 29. The time constants of the peak of the m-variable for both the K+ delayed rectifier and of the M current should be in seconds not in mV. 

Reviewer #2, Andrea Nistri, SISSA, Trieste: The MS by Verneuil et al. reports an interesting study of the role of the M current (IM) in the operation of the spinal networks responsible for locomotion. These data are novel and of general interest as they indicate important molecular and biophysical components of a circuitry that relays on the interplay amongst a number of ionic conductances expressed by key neurons to generate walking. The present study is multidisciplinary and well organized and requires a relatively small degree of attention to improve it.

Although the MS is overall clearly written, the authors are advised to improve the English style and to proofread the text in order to eliminate editing errors such as having names of quoted references alongside the ref. number.

It is suggested that the editor should check if there is a suitable repository for the original experimental data.

The authors state that "a decrease of IM converts a spiking pattern into a bursting mode in a subset of ventromedial spinal interneurons" (p. 15). How would this occur physiologically? What cells would release ACh? Information on this subject would be useful.

The authors report that injection of retigabine increased the speed of rat walking behaviour. However, retigabine was used as a second line antiepileptic drug (now discontinued) and there are no reports of a similar effect on man.

Since the patch pipette contained K-gluconate, it is important to indicate if the membrane potential values were corrected for junction potential.

The Methods section should contain more details about the antibody specificity with suitable references (western blotting, KO animals etc).

The argument that TEA does not affect the M current should be thoroughly revised. There are numerous reports that even at 1 mM concentration TEA can inhibit this current. See for example: Freschi JE. J Neurophysiol. 1983;50:1460-78; Constanti A, Sim JA. J Physiol. 1987;387:173-94; Halliwell JV, Horne AL. J Physiol. 1995;487:421-40. 

Importantly, previous studies have shown that the subunit composition of the M current determines the sensitivity to TEA. See Smith JS et al. J Neurosci. 2001;21:1096-103, and Shah M et al. J Physiol. 2002,544:29-37. These reports should be discussed in the light of the MS results.

In general, the actual amplitude of the M current is not very large. It would be interesting if the authors could discuss a bit more the relative contribution of this conductance versus other K currents in determining bursting patterns.

Reviewer #3: General Comments

This is an interesting paper by an excellent group on the interaction between Kv7 channels and persistent sodium currents. The group has published several findings on the role of persistent inward currents and their role in locomotor activity. This is the first paper to examine the interesting interactions that occur between currents that appear to operate together in regards to burst dynamics. They first show the effects of M-current manipulation in vivo and then through a series of whole-cell patch, modelling, and in vitro fictive locomotion studies show the effects of M-currents on locomotion. Interestingly, they show a rather dramatic interaction between IM and INaP where IM works in opposition to INaP to control the duration of bursts and the frequency of pacemaker cells. So overall excellent work uncovering a new mechanism for burst regulation within the locomotor central pattern generator of the spinal cord. 

Major Comments

I was pleased to see the authors show that M-currents contribute to locomotion in vivo, but some issues regarding interpretation remain. The authors acknowledge that manipulation of M-currents using IP injections would affect locomotor centres throughout the brain. What would be useful here, if possible, would be to run a set of experiments using intrathecal injections targeting the L1/2 segments of the spinal cord with the agonists/antagonists. That said I was impressed that the fictive locomotor activity showed responses that were similar to the in vivo situation. 

Minor Comments

Results

Page 5. Line 20. I think the summary statement that Kv72/3 channels are affecting the clock is something that is best discussed rather that concluded at the early stage of results. There are other possibilities here for fast/slow walking that don't include the clock but could include changes in descending drive. 

Page 6. Line 24. Add a citation to the 'electrophysiological signature of IM'.

Figure 1. Exchange ICA73 and Linopiridine traces in A. That would put the agonist/antagonists pairs in the same row. 

Figure legends. Make it clear species in legends. Fig 3 has both rat and mouse. 

Discussion

I would have a separate paragraph devoted to the paper unravelling the paradox of how increasing IM which decreases excitability translates to an increase in the rhythmic frequency of walking. 

Page 15 - line 23. Plead to suggest. 

Page 17. Lines 19-20. Reword.

Page 18. Line 19. Aside from.

Methods. 

Throughout. It is not very clear why the concentrations of agonists/antagonists used were chosen. 

Reviewer #4: The manuscript by Verneuil et al attempts to define the function of the M-current and KCNQ channels in locomotion CPG, an unexplored and important topic. They authors use an array of tools including a Kcnq2 knockin mouse model of a recurrent pathogenic mutant (KCNQ2T274M) to determine the precise function of KCNQ channels and its relationship to the sodium persistent current in spinal cord interneurons. This work strongly suggests that the M-current regulates the activity of the sodium persistent current, which in turn controls the firing behavior of locomotion interneurons and locomotion CPG in general. Although this work provides a significant advance in our understanding of KCNQ channels in the nervous system, some of the conclusions need to be better flushed out by the authors.

Major Comments

(1) It is well established that the molecular correlate of the M-current is KCNQ2/3 channels. However, over the years there have been suggestions that KCNQ2 homomeric channels could mediate the M-current. Based on immunostaining of ventromedial interneurons the authors postulate that the M-current in spinal interneurons is likely due to axonal KCNQ2 channels. Unfortunately, the data do not fully support this assertion. First, in Figure 2 there is a substantial signal for both KCNQ2 and KCNQ3 at the soma. The signal is everywhere in the soma. Somatic signal was not quantified. Two, their conclusion is based on non-validated antibodies. Three, 100nM retigabine (Fig S3) increased the holding current and amplitude of the M-current. Retigabine has very high preference for KCNQ3 and KCNQ2/3 channels, over KCNQ2 channels, particularly at 100nM (Schenzer et al 2005 JNS Figure 1). Fourth, the use of KCNQ2 T274M mice suggest that KCNQ2 channels mediate the M-current, but does not exclude the possibility that KCNQ2/3 also mediates it as KCNQ2T274M will also reduce KCNQ2/3 currents.

Thus, to better examine whether the KCNQ2 or KCNQ2/3 channels mediate the M-current in spinal interneurons the authors need to test the effect of 5mM TEA on the M-current. 5 mM TEA fully blocks KCNQ2 homomers but only partially KCNQ2/3 channels. Please see Carver et al (Figure 4A JNS 2019) for similar experiments in neurons. 

2. The M-current values only reflect a fraction of the total KCNQ mediated currents in neurons. The authors, rather than reporting only M-current values (deactivating component), they should also report the total XE991 sensitive current that includes the faster KCNQ2/3 current component in Figure 3. Additionally, the authors should also show the re-activating component of the current (the return step). 

3. The G-V shifts using the ICA73 are difficult to understand (Figure 3). They authors show that application of ICA73 increases the holding current, likely, due to opening of KCNQ2 mediated M-current. This is consistent with the established ICA73 mechanism of action, which is to cause a left shift to the KCNQ2 channels G-V (Wang et al 2018 JGP). However, the authors do not see a significant shift in the G-V. Has the ICA application alter kinetics of the M-current as it usually does (Wang et al 2018 JGP)? Please clarify why ICA73 does not shift the G-V to hyperpolarized Vm.

4. The authors did not find any effect of XE991 in the spiking activity of ventromedial interneurons. The attributed this to the poor blocking effect of XE991 at resting membrane potentials. This poor blocking issue can be bypassed by prolonging the application time of XE991 as shown by Hu and Bean (Neuron 2018). Therefore, the authors should repeat the experiments in Figure 4B using a 30 minute application of XE991.

5. Because of the lack of effect of XE991 in ventromedial interneurons the authors chose to use modeling. It would have been better to use the Kcnq2 T274M mice. Consequently, the authors need to report whether the firing properties of ventromedial interneurons are increased in Kcnq2T274M mice. This will strengthen their conclusions over modeling, which is based on several assumptions.

6. The key experiment in this manuscript is to show whether the sodium persistent current has increased in Kcnq2T274M mice and whether Kcnq2T274M mice have gait or locomotion coordination issues as shown in Figure 1 using pharmacology.

Minor comments.

1. The title does not convey clearly the conclusions of the paper. It is more appropriate for a commentary.

2. The abstract needs to be revised and remove the statement about axonal KCNQ2 channels. The work does not distinguish between the roles of axonal to somatic KCNQ2 channels.

3. For statistics the authors need to report number of cells, slices, and number of animals used for each experiment.

---

## [Decision Letter · Decision Letter 2]

21 Sep 2020

Dear Frédéric,

Thank you for submitting your revised Research Article entitled "It takes two to tango: M-current swings with the persistent sodium current to set the speed of locomotion." for publication in PLOS Biology. I have now obtained advice from the original reviewers and have discussed their comments with the Academic Editor. You will note that reviewers 1 and 2, Tuan Vu Bui and Andrea Nistri, have revealed their indentities.

We're delighted to let you know that we're now editorially satisfied with your manuscript. However before we can formally accept your paper and consider it "in press", we also need to ensure that your article conforms to our guidelines. A member of our team will be in touch shortly with a set of requests. As we can't proceed until these requirements are met, your swift response will help prevent delays to publication. Please also make sure to address the data and other policy-related requests noted at the end of this email.

To submit your revision, please go to https://www.editorialmanager.com/pbiology/ and log in as an Author. Click the link labelled 'Submissions Needing Revision' to find your submission record. 

*Copyediting*

*Published Peer Review History*

*Early Version*

Sincerely,

Gabriel Gasque, Ph.D.,

Senior Editor,

ggasque@plos.org,

PLOS Biology

DATA POLICY:

-- Please relabel your Data Files as S1 Data, S2 Data, etc, verbatim, and adjust your manuscript accordingly. 

-- Please re-label the tabs of your Data File for Figures 5, 6, 7, 8, S4, S5, S6, S7, S8, and S9 since they are mislabeled. 

-- Please ensure that your supplemental data files have a legend.

Reviewer remarks:

Reviewer #1, Tuan Vu Bui: The authors have addressed my concerns in a satisfactory manner. While I leave it to the other reviewers to assess how their concerns were addressed, the additional experiments seem to strengthen the conclusions of the paper with regards to the role of M-type K+ current to the operation of spinal locomotor circuits.

Reviewer #2, Andrea Nistri: The authors have satisfactorily addressed all the concerns expressed by the reviewers and have added substantial new data to the MS. There is no doubt the MS is largely strengthened and in line with the PLOS Biology standards.

Reviewer #3: I have examined the authors responses to my concerns. I was particularly pleased to see that they tackled the intrathecal administration of drugs in vivo. These are not easy experiments and they have strengthened the manuscript. My other concerns were addressed. 

Reviewer #4: This is a revised manuscript defining the role of the M-current in locomotion CPG using a multi-disciplinary approach. The authors have addressed all my previous concerns with the additions of new experiments. This is a very well done and rigorous study that would be of interest to many scientists of multiple disciplines. I have no additional comments.

---

## [Editor Report · Decision Letter 3]

13 Oct 2020

Dear Dr Brocard,

On behalf of my colleagues and the Academic Editor, Frederic Bretzner, I am pleased to inform you that we will be delighted to publish your Research Article in PLOS Biology. 

Early Version

PRESS 

Kind regards,

Alice Musson

Publishing Editor, 

PLOS Biology

on behalf of

Gabriel Gasque,

Senior Editor

PLOS Biology